
# A multi-ice-core, annual-layer-counted Greenland ice-core chronology for the last 3800 years: GICC21

Giulia Sinnl[1], Mai Winstrup[2], Tobias Erhardt[3,4], Eliza Cook[1], Camilla Jensen[4], Anders Svensson[1], Bo Møllesøe Vinther[1], Raimund Muscheler[5], and Sune Olander Rasmussen[1]

[1] Physics of Ice, Climate, and Earth, Niels Bohr Institute, University of Copenhagen, Denmark.

[2] DTU Space, National Space Institute, Technical University of Denmark, Kongens Lyngby, Denmark

[3] Alfred Wegener Institute, Helmholtz Centre for Polar and Marine Research, Bremerhaven, Germany

[4] Climate and Environmental Physics & Oeschger Center for Climate Change Research, Physics Institute, University of Bern, Switzerland

[5] Quaternary Sciences, Department of Geology, Lund University, Sweden

**Correspondence:** Giulia Sinnl (giulia.sinnl@nbi.ku.dk)

**Abstract**

Ice-core timescales are vital for the understanding of past climate; hence they should be updated whenever significant amounts of new data can contribute to improvements. Here, the Greenland ice-core chronology was revised for the last 3835 years by synchronizing six deep ice-cores and three shallow ice-cores from the central Greenland ice sheet. A layer-counting bias was found in all ice cores because of site-specific signal disturbances, and a manual comparison of all ice cores was deemed necessary to increase timescale accuracy. A new method was applied by combining automated counting of annual layers on multiple parallel proxies and manual fine-tuning. After examining sources of error and their correlation lengths, the uncertainty rate was quantified to be one year per century.

The new timescale is younger than the previous Greenland chronology by about 13 years at 3800 years ago. The most recent 800 years are largely unaffected by the revision, while the slope of the offset between timescales is steepest between 800 and 1000 years ago. Moreover, offset-oscillations of about 5 years around the average are observed between 2500 and 3800 years ago. The non-linear offset behavior is attributed to previous mismatches of volcanic eruptions, to the much more extensive data set available to this study, and to the finer resolution of the new ice-core matching.

In response to volcanic eruptions, averaged water isotopes and layer thicknesses from Greenland ice cores provide evidence of notable cooling lasting for up to a decade, longer than reported in previous studies of volcanic forcing. By analysis of the common variations of cosmogenic radionuclides, the new ice-core timescale is found to be in alignment with the IntCal20 curve. Radiocarbon dated evidence found in the proximity of eruption sites such as Vesuvius or Thera was compared to the ice-core dataset; no conclusive evidence was found regarding if these two eruptions can be matched to acidity spikes in the ice cores. A hitherto unidentified cooling event in the ice cores is observed at about 3600 years ago (1600 BCE), which could have been caused by a large eruption which is, however, not clearly recorded in the acidity signal. The hunt for clear signs of the Thera eruption in Greenland ice-cores thus remains elusive.

## 1. Introduction

The Late Holocene saw the development of the entire human civilization, while also being characterized by a stable climate punctuated by short-term events, such as volcanic eruptions. While it is certain that the impact of climate closely influenced human history, open questions remain about how climatic shifts happened across regions and when climate change has been most catastrophic. Fortunately, many detailed regional proxy records cover this period and can help reconstruct the history of recent climate.

Paleoclimatic chronologies allow the synchronization of such proxy records from different geographic locations, thereby providing a fundamental tool for the understanding of the Earth's climate. One method for timescale reconstructions is annual layer counting, which requires a well-preserved archive of seasonal climate variability from, for example, polar ice cores, tree ring chronologies, or sedimentary data from marine and terrestrial samples (Stuiver & Pearson, 1986; Brauer, Endres, & Negendank, 1999; Andersen et al., 2006). Other timescale construction techniques include depth-age modelling (Johnsen et al., 2001) or radiometric dating methods, such as carbon dating, which needs correction by radiocarbon age


calibration (Reimer et al., 2020). Once the timescales from different geographic locations are set up and precisely synchronized, it may be possible to investigate lead-lag dynamics within the broader climate system. In the Holocene, recent studies investigated the comprehensive impact of volcanic eruptions, suggesting for example a 10-year cooling in European summer temperatures (Sigl et al., 2015) or a 5-year positive North Atlantic Oscillation (NAO) (Sjolte et al., 2018), both after tropical eruptions. Other examples of such inter-regional comparative studies dependent on timescale accuracy





are the study of bi-polar timing of climate changes in the last glacial (WAIS Members, 2015; Pedro et al., 2018; Svensson et al., 2020) or the timing of the Holocene onset over Greenland and Asia (Nakagawa et al., 2021).

### 1.1. Annual layers in Greenlandic ice cores

Ice cores from Greenland contain high-quality climatic information thanks to the steady deposition of snow and impurities and to the inclusion of air bubbles in the ice, which have occurred continuously since the formation of the ice sheet. The
deposited snow is enriched by a variety of chemical compounds, such as sodium, calcium, ammonium ions, and water-insoluble particles, like dust and volcanic ashes, which may all be interpreted as proxies for climatic conditions and processes (Fuhrer, Wolff, & Johnsen, 1999; Rhodes, Yang, & Wolff, 2018). Moreover, the isotopic composition of the deposited snow stands as a proxy for temperature and moisture at the drill-sites (Dansgaard, 1964; Johnsen, Dansgaard, & White, 1989).

The quality of the retrieved signals is highest during high-accumulation periods and especially at high-accumulation sites, since for example isotopes are heavily affected by diffusion (Johnsen, 1977). In the Holocene period (last 11.7 ka) snow deposition was relatively high and stable. Furthermore, the Holocene ice roughly comprises the upper half of the central Greenland ice sheet and is not affected by ice thinning at the same level as the older, much thinner, glacial layers (Vinther et al., 2009; Gkinis et al., 2014; Gerber et al., 2021). However, the data quality in the Holocene is hampered by the brittle
ice zone, which is found at depths at which high-pressure gas bubbles in the ice make the core very fragile (Neff, 2017).

Some proxies follow a clear annual cycle that can be observed if the annual layer thickness and the analytical method provide sufficient resolution. The seasonal patterns of ice-core proxies are determined by complex depositional dynamics that control the transport from the sources to the ice sheet (Gfeller et al., 2014, Whitlow et al., 1992, Beer et al., 1991, Fischer et al., 1998, Fuhrer et al., 1996). Sodium ($Na^{2+}$) has a strong winter peak because of increased advection of marine
air masses (Herron, 1982); calcium ($Ca^{2+}$) peaks in the spring because of enhanced transport from terrestrial reservoirs (Whitlow et al., 1992); ammonium ($NH_4^+$) has a maximum in the late spring/summer because of enhanced biogenic activity in the North American continent (Fischer et al., 2015); nitrate ($NO_3^-$), which is also related to biogenic processes, peaks in the summer (Herron, 1982; Röthlisberger et al., 2002). Water isotopes ($\delta^{18}O$ and $\delta D$) show a sinusoidal pattern with winter valleys and summer peaks, mainly representing temperature variations at the drill site (Jouzel et al., 1997).

### 1.2. Annual layer counting methods

Thanks to the proxies that show an annual pattern, annual layers in ice cores can be counted manually, a process that has always been a challenging part of ice-core timescale reconstructions (Vinther et al., 2006, referred to as Vinther06 in the rest of this article; Rasmussen et al., 2006; Sigl et al., 2013). The counting can be done on continuously measured water isotopes, before diffusion becomes too heavy, or on soluble ions, such as sodium, calcium, and ammonium. To correct for
the diffusion, it can be necessary to apply deconvolution techniques to reconstruct the original annual layers (Vinther06).

Manual identification of annual layers is a time-consuming and inherently subjective task, and attempts have been made to automate the process (McGwire et al., 2008; Smith et al., 2009). StratiCounter (SC) is a software package that computes the most likely sequence of annual layers in an ice-core multi-proxy dataset (Winstrup M., 2011; Winstrup M. et al., 2012). Starting from example-data provided by the user and applying a Hidden Semi Markov Model, the algorithm learns to
recognize the specific annual pattern. SC provides a layer count and a probability distribution of the recognized layer boundaries. Some initial settings determine if the program should, for example, reduce the resolution of the original data, apply some pre-processing, or give different weight to the different data series in the analysis. These requirements are both ice-core and proxy dependent.

### 1.3. Holocene stratigraphic markers

Short-term events, such as volcanic eruptions or biomass burning events, may be used to synchronize ice cores if the corresponding layers can be unambiguously seen in several ice cores. Volcanic eruptions constitute the most robust base for the matching of ice core timescales because they leave a clear imprint in the ice-core signal. When available, sulfate ($SO_4^{2-}$) measurements are used to locate individual eruptions, because of the emission of sulfur compounds to the atmosphere that precipitate onto the ice sheet (Lin et al., 2021). Eruptions are also recorded as prominent peaks in the
Electrical Conductivity Measurements (ECM) and in the dielectric permittivity (DEP) (Clausen et al., 1997; Mojtabavi et al., 2020; Hammer, 1980; Wilhelms et al., 1998). The volcanic-eruption signal usually spans more than one year, so that one can identify the start, the maximum, and the end of the event (Clausen et al., 1997).

When volcanic ash layers (tephra) are found in ice cores and geochemically and stratigraphically matched to reference deposits from the origin site, the identity of the volcano can be confirmed (Davies et al., 2010; Abbott and Davies, 2012;
Bourne et al., 2014; Cook et al., 2018a). Otherwise, if the source volcano has not yet been found, geochemical similarity of layers found in different ice cores provides evidence of synchronicity (Cook et al., 2018b). However, for the most part, there is no tephra associated with acidity peaks of assumed volcanic origin, and thus matching of the cores relies entirely on identification of corresponding patterns of acidity peaks.



The chemo-stratigraphic response to volcanic eruptions can vary between ice cores due to different depositional dynamics. The recorded shape and delay of the volcanic signal depends, for example, on the distance from the eruption site, on the balance between dry and wet deposition of sulfate, on snow redistribution, or on different deposition levels at the ice core site (Robock and Free, 1995, and references therein; Gautier et al., 2016). Therefore, it can happen that a very strong eruption signal at GRIP from e.g. an Alaskan eruption will only appear as a minor signal in the DYE-3 core, because the two drilling sites have a different distance from the origin and, moreover, receive snowfall during different meteorological situations (Clausen et al., 1997). Hence the matching cannot rely only on the similarity between single peaks, but must also depend on the patterns they form in. Still, the link between many historical eruptions and the corresponding ice-core acidity spikes is well-established and serves as an exact time reference (Sigl et al., 2015, referred to as Sigl15 in the rest of this paper).

Ammonium ($NH_4^+$) is a proxy for biogenic activity (Fuhrer et al., 1996), and ammonium spikes have been directly linked to biomass burning events, i.e. wildfires (Fischer et al., 2015). It is not possible to find the origin of wildfires with the same certainty as for volcanic eruptions, but patterns of ammonium-rich years can nonetheless be identified across the Greenland ice cores, providing additional tools for synchronization. Because of the alkaline nature of $NH_4^+$, the ECM will record marked dips (Rasmussen et al., 2006). To some degree, nitrate ($NO_3^-$) peaks are sometimes observed to coincide with $NH_4^+$ spikes.

Other events that serve as matching points include variability of cosmogenic radionuclides, which are caused by solar storms or by other forms of solar variability (Muscheler, Adolphi, & Knudsen, 2014). By measuring the co-registration of two tie-points such as the 775 CE and the 994 CE events, Sigl15 showed that Beryllium-10 ([10]Be) enhancements provide robust constraints of alignment between tree-ring and ice-core timescales. In the recent work by O´Hare et al. (2019), the signature of an intense solar storm was identified in ice cores, at an age of about 660 BCE, which provides an added alignment point between ice cores and tree-ring timescales in the late Holocene.

### 1.4. The GICC05 timescale in the Holocene

The Greenland Ice Core Chronology 2005 (GICC05) is the most widely recognized timescale for Greenland ice-core studies (Vinther06). In the Holocene, three ice cores form the basis of GICC05: the DYE-3 ice core from southern Greenland (Johnsen et al., 2001), the GRIP ice core from Summit/central Greenland (Dansgaard et al., 1993), and the NorthGRIP ice core from north-western Greenland (NGRIP members, 2004). The ice cores were matched by recognizing common volcanic eruptions in the ECM signal, and the annual layers were manually counted using water isotopes ($\delta^{18}O$ and $\delta D$ are available in overlapping sections and are equally suited for annual layer identification). In the older part of the Holocene, high-resolution impurity records were also included in the layer counting, but they were not available at the time for reconstructing the timescale for the Late Holocene (Rasmussen et al., 2006).

When the ice-core data quality was not equal between the three cores, a master chronology was produced on the best resolved record, which was then transferred to the other ice cores. Until 1813 years b2k (years before 2000 CE, same convention applied in the rest of this paper), the count was produced on isotopes from DYE-3 and deconvoluted isotopes from GRIP and NorthGRIP. From 1813 until 3835 years b2k, NorthGRIP ages were transferred from DYE-3 and GRIP, because of lacking isotope data.

In the construction of GICC05, the Vesuvius eruption (79 CE, Italy) was considered an exact time-marker carrying no age uncertainty, based on a tephra deposit found in NorthGRIP (Vinther06; Barbante et al., 2013). Recent analysis has however confirmed that the sample most likely originates from an Alaskan eruption (Plunkett et al., 2021), thus leaving the chronology with a significant synchronization weakness. Recent comparisons between Greenland and [14]C chronologies have exposed issues with the layer count in selected sections of the timescale (Muscheler et al., 2014; Sigl15; Adolphi & Muscheler, 2016; McAneney & Baillie, 2019), mainly caused by Vesuvius being attributed wrongly. As another discussion point, Hammer (1980) and Vinther et al. (2006) attributed the massive Hekla 1104 CE eruption to a prominent ECM peak in GICC05. Later, robust geochemical analysis of tephra grains found close to the peak disproved the Hekla origin, and sulfate-isotope measurements suggested that the eruption was bi-polar, rare for Icelandic eruptions (Coulter et al., 2012; Sigl15). In this work we aim to expand on the causes of the GICC05 mismatches and to investigate on possible other problems to be resolved, such as, for example, the robust assessment of ice core timescales uncertainties.

### 1.4.1. Uncertainty estimates of GICC05 in the Holocene

The uncertainty associated with a timescale provides essential information for a correct interpretation of the climatic data. The most important source of uncertainty in GICC05 was considered to be the misinterpretation of annual layers by the observers. By defining uncertain layers to be features in the ice core that could neither be dismissed nor confirmed as annual layers (Vinther06), the GICC05-uncertainty was estimated from the Maximum Counting Error (MCE), that is, half the sum of the uncertain layers accumulated until the corresponding age (Rasmussen et al., 2006). Thus, each uncertain layer contributes with ½ ± ½ years to the age scale, whereas certain layers contribute 1 ± 0 years.





A fundamental choice in uncertainty estimation is whether one assumes correlation between errors. In case the errors are assumed uncorrelated, then they should be summed in quadrature. If, on the other hand, all errors are assumed to be fully correlated, then the total uncertainty at a given time is a linear sum of the individual errors. Acknowledging that for the case of ice cores the errors are likely neither fully correlated nor uncorrelated, the authors of GICC05 opted for a conservative approach and summed the MCE linearly, but in turn did not include contributions from other sources than misinterpretation of annual layers.

Still, GICC05 was considered exact for the part younger than the Vesuvius eruption, because many well-known historical eruptions tied the chronology together. Therefore, the published uncertainty until about 2.7 ka b2k is only 2 years, while the published GICC05 uncertainty at 3.9 ka b2k is 5 years.

The MCE does not account for bias in the counting process. To estimate a possible observation bias, the authors observed that the count between 1362 CE (Öræfajökull, Iceland) and 79 CE (Vesuvius) was correct within one year, corresponding to ~0.1% of the interval. As this number was smaller than the MCE it was considered negligible.

### 1.5. The NS1-2011 timescale

A more recent Greenland ice-core timescale for the past 2500 years was based on new bipolar tie-points, such as volcanic tephra and solar storm data, and on new high-resolution multi-parameter impurity records (Sigl15). This timescale will be referred to as the NS1-2011 chronology. StratiCounter was employed to count annual layers on the NEEM-2011-S1 shallow ice core (Sigl et al., 2013) and on the NEEM main/deep ice core (Sigl15). These ice cores were matched to NorthGRIP via 15volcanic and solar markers to allow for comparison to GICC05. Moreover, a manual count on the NEEM main core was conducted for the oldest ~500 years of the chronology, which are not covered by NEEM-2011-S1 dataset.  For most of the timescale, SC was run in constrained mode using volcanic tie-points of known age. The earliest exact time marker applied for the chronology is the 536 CE eruption, prominent in the acidity and sulfate records (Sigl15). For ice older than 536 CE, the authors analyzed detailed records of historical, literary, and climatic evidence and found that the timescale aligns with most of the validation points, providing statistical tests to evaluate the significance of the result. The tests were repeated with GICC05 and the results were not significant, demonstrating the superior accuracy of the NS1-2011 timescale.

The NS1-2011 offset to GICC05 at around 79 CE was quantified to be 8 years and the age of the layer formerly attributed to Vesuvius was changed to 87/88 CE.

The uncertainty estimate of the NS1-2011 timescale was based on the SC probability estimate. The age of volcanic eruptions is reported as a weighted average of the SC-counts in NEEM-2011-S1, the NEEM main core, and WDC. For example, the age of the Indonesian Samalas eruption (Vidal et al., 2016) is given as 1258 ± 2 years CE. Moreover, the comparison between the manual and the automated count in NEEM amounted to a difference of 1 year over the 500 year time interval. The timescale was estimated to have a 5 year uncertainty at 2500 years b2k.

### 1.6. The need for a revised and unified Greenland ice-core chronology in the Holocene

Given the known inconsistencies between existing Greenland Holocene timescales and thanks to the advent of a new high-resolution dataset from the recent EastGRIP ice core (Mojtabavi et al., 2020; Erhardt et al., In Prep.), we find it timely to revise the GICC05 timescale, providing a new unified ice-core chronology that includes all available data from Greenland deep ice cores. Our method relies on parallel dating of multiple cores with well-resolved annual data covering the period back to 3835 years b2k, which ensures data coverage from at least four ice cores until the brittle ice zone affects the ice-core data quality.

SC cannot presently be applied to multiple ice cores together. Nevertheless, SC can be applied separately to each ice-core, on its own depth scale, after which the resulting counts can be combined between cores. Hence, SC cannot at this time provide a fully automated multi-core timescale. Furthermore, SC cannot be used to assess whether the ice-core signal is affected by disturbances that might have altered its shape, such as snow redistribution, melt layers, and multiple seasonal peaks in the proxies (Mosley-Thompson et al., 2001; Westhoff et al., 2021; Lei Geng et al., 2014). These observations rely on a comparison of records from several ice cores. Hence, an extensive manual effort is still required to identify problematic layers and to bring a multi-ice-core timescale to a final state.

We find the MCE not suited to our timescale since it is a single-record uncertainty estimate that does not capture the complexity of the multi-core chronology: uncertain layers in one core may be certain in another, and a comparison of the two can solve many chronological issues. Based on the combination of statistical estimates and empirical observation, we propose a simple formula to provide the new timescale, named GICC21, with a robust, consistent, and user-friendly uncertainty estimation. Furthermore, thanks to a good geographical coverage of the central Greenlandic ice sheet provided by our dataset, our improved synchronization of Greenlandic ice cores allows more precise investigations of the relative timing of climatic events, such as the climatic response to volcanic eruptions as reflected in the ice-core water isotopic climate proxy.


The oldest part of GICC21 includes the likely age range of the intensely studied Thera eruption from Santorini (Greece). Archaeological evidence generally places the eruption in the 15th or late 16th century BCE (Manning et al., 2015), but carbon-dated evidence found at the eruption site (Ramsey et al., 2004; Friedrich et al., 2006), and calibrated through IntCal13 (Reimer et al., 2013), is at odds with this age range, preferring an older age within the late 17th century BCE. However, the

calibrated [14]C age ranges for the eruption have been updated through renewed tree-ring counted sections (Pearson et al., 2018; Friedrich et al., 2020), which showed an offset of 24 years towards younger ages of the previously identified age range. The consequences of these studies reveal that the probabilities for a Thera eruption in the 16th century BCE (3500-3600 years b2k) have increased. As a consequence of these and other chronological issues, the IntCal13 calibration curve has been updated to the IntCal20 curve, as new data became available (Reimer et al., 2020). Furthermore, Pearson et al.

(2020) provided evidence of an interesting calcium anomaly in the tree-samples close to the eruption site (Turkey) at 1560 BC (3560 years b2k), found by pattern-matching of parallel tree-ring records. They speculated that this might in fact be a new possible candidate for the Thera eruption age, since calcium depletion in trees indicates a response to drought or forest fires.

From the ice core-perspective, Hammer et al. (1987) and Vinther06 proposed a 17th century BCE ice-core age for Thera

based on the presence of tephra grains. The claim was disputed (Denton and Pearce et al., 2008) based on improved geochemical matching of the ice core tephra to eruption from Aniakchak, Alaska (Pearce, Westgate, Preece, Eastwood, & Perkins, 2004). By improving ice-core chronologies until the Thera age range, we intend to apply GICC21 to consolidate analogies between radiocarbon dated samples and ice-core data.

## 2. Data

Data from 6 deep and 3 shallow ice cores provide the basis for GICC21; details about each drilling site are given in Table 1. The resolution and the quality of the data reflect not only the local climatic conditions but also the state of the technology at the time of retrieval and measurement. Moreover, data is only available, or of sufficient resolution and quality for layer counting, at selected ice-core depth ranges, as can be seen in Figure 1b.

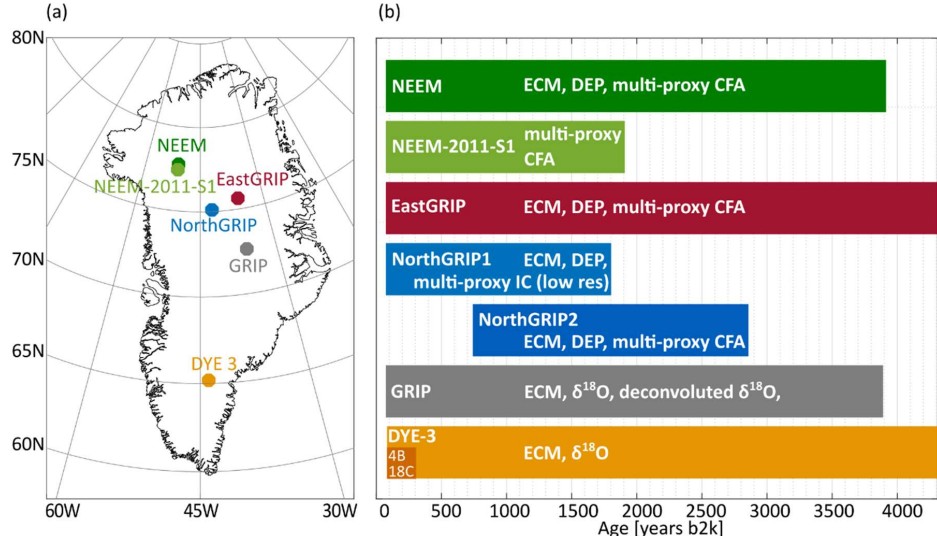

Figure 1 Overview of the data used for this study. (a) Geographic locations of the ice cores, of which Table 1 contains the site specifications. (b) The colored patches summarize the available datasets used for annual-layer counting and inter-core matching, plotted on their approximate age-range.



Table 1 Specifications about the ice cores included in this study, ordered by coring location, North to South.

| Ice Core | Elevation, m | Lat., °N | Long., °W | Mean Air Temp., °C | Accumulation, m ice per year | Length, m | Years of Drilling | Brittle ice zone, m |
|---|---|---|---|---|---|---|---|---|
| NEEM-2011-S1 | 2450 | 77.45 | 51.06 | -21[c] | 0.22[d] | 410 | 2011[d] | |
| NEEM | 2479 | 77.25 | 51.09 | -29[a] | 0.22[a] | 2540 | 2008–2012[b] | 609-1281[k] |
| EastGRIP | 2458 | 75.38 | 36.00 | -29[*] | 0.12[e] | 2150 | 2017–2019 (ongoing) | 650-950[l] |
| NorthGRIP1 | 2917 | 75.10 | 42.32 | -32[f] | 0.19[f] | 1351 | 1996–1997[g] | 790-1200[f] |
| NorthGRIP2 | 2921 | 75.10 | 42.32 | -32[f] | 0.19[f] | 3085 | 1997–2004[g] | 790-1200[f] |
| GRIP | 3230 | 72.58 | 37.64 | -32[f] | 0.23[f] | 3027 | 1989–1992[h] | 800-1300[f] |
| DYE-3 | 2480 | 65.18 | 43.83 | -20[f] | 0.56[i] | 2037 | 1979–1981[j] | 800-1200[f] |
| DYE-3 4B | 2491 | 65.17 | 43.93 | -20[*] | 0.535[f] | 174 | 1983[j] | |
| DYE-3 18C | 2620 | 65.03 | 44.39 | -20[*] | 0.44[f] | 113 | 1984[j] | |

[a] (NEEM community members, 2013)[b] (Rasmussen et al., 2013)[c] (Faïn, et al., 2014)[d] (Sigl, et al., 2013)[e] (Gerber, et al., 2021)[f] (Vinther, et al., 2010) [g] (Dahl-Jensen et al., 2002) [h] (Johnsen et al., 1992) [i] (Vinther06) [j] (Clausen et al., 1988)[k](Warming et al., 2013) [l](Westhoff et al., 2021)[*]estimated from PROMICE data (Ahlstrøm et al., 2008) [*]Assumed same as DYE-3

### 2.1. EastGRIP

The East GReenland Ice-core Project (EastGRIP) is an ongoing drilling effort which in the latest field season (2019) reached a depth of about 2150 m. The ECM and DEP measurements were made in the field camp on 1.65 m long pieces of ice (Mojtabavi et al., 2020). Chemistry records of aerosol impurities were measured in Bern using the proven Continuous Flow Analysis (CFA) Setup (Kaufmann et al. 2008) coupled to an ICP-TOFMS (Erhardt, Jensen, Borovinskaya, & Fischer, 2019). The chemistry data include a vast range of different species measured continuously at high sampling resolution (1 mm), with a resolving power of about 1 cm, making this dataset among the most detailed available, however the annual accumulation rate is also the lowest of the records included (Table 1). A detailed description of the setup and the data used in this study can be found in Erhardt et al. (in prep). For counting layers, we used Na, Ca, $NH_4^+$, and $NO_3^-$ concentrations from 13.82 to 460.30 m depth. Water isotope records were also measured continuously but, due to the low annual accumulation, the annual signal does not survive diffusion in the firn, and therefore cannot be used for annual layer identification.

### 2.2. NorthGRIP

The NorthGRIP drilling was completed in 2004 and is composed of two ice cores: NorthGRIP1 and NorthGRIP2 (Dahl-Jensen et al., 2002). For NorthGRIP1, ECM data are available until the core ends, at about 1351 m, and discrete chemical measurements (5 cm resolution) are available uninterruptedly until 350 m and in short fragments below this (Vinther06). Despite the resolution of only 4-5 samples per year, the annual layer pattern is clearly recognizable, and we used ECM, $Na^+$, $Ca^{2+}$, $NH_4^+$, $NO_3^-$, $Cl^-$, $Mg^{2+}$, $SO_4^{2-}$, and $\delta^{18}O$ for layer counting between 9.9 m and 349.1 m depth.

For NorthGRIP2, a chemistry dataset was measured at the Desert Research Institute (DRI) with a resolution of 1 cm (McConnell et al., 2018). We used Na, Ca, $NH_4^+$, and $NO_3^-$ for layer counting between 159.6 m to 582.4 m depth. For volcanic matching, we mainly used the DEP.

### 2.3. NEEM and NEEM-2011-S1

The North Greenland Eemian Ice Drilling (NEEM) was completed in 2012 (NEEM community members, 2013). ECM and DEP were measured in the field (Rasmussen et al., 2013). Impurity records, measured by an international team coordinated by the University of Bern, were as well obtained in the field and have a depth resolution similar to the EastGRIP data set (Kaufmann et al., 2008) but, due to brittle ice, suffer from increasingly wide data gaps that make annual-layer identification difficult below around 750 m. Hence, we used ECM, $Na^+$, $Ca^{2+}$, $NH_4^+$, and $NO_3^-$ for layer counting between 7.6 m and 727.3 m depth. A detailed description of the NEEM CFA measurements and the dataset can be found in Erhardt et al. (in review).

The NEEM-2011-S1 ice core is a 410 m shallow core that was drilled about 100 m away from the NEEM main core (Sigl et al., 2013). This core reaches back until the volcanic layer attributed in GICC05 to the Vesuvius eruption (79 CE). Chemistry data was measured at DRI for the entire core length, but the ECM of the shallow core was not measured, hence we relied on non-sea-salt sulfate (nss-S) for volcanic matching (Sigl et al., 2013). The species Na, nss-Ca, nss-Na, $NH_4^+$, $NO_3^-$, and BC (Black Carbon) were used for layer counting between 6.1 m and 410.8 m, in order to achieve a similar SC-output as in Sigl15.



### 2.4. GRIP

The drilling of the GRIP ice core (or Summit ice core) was completed in 1992 (Johnsen et al., 1992). Only ECM and isotope data are available for counting layers in this core in the late Holocene. The annual signal in the isotope data is moderately affected by diffusion, but deconvolution restores a very strong sinusoidal pattern that can be used for annual counting (Johnsen et al., 2000; Vinther, Johnsen, Andersen, Clausen, & Hansen, 2003). Hence, we used ECM, $\delta^{18}O$, and deconvoluted $\delta^{18}O$ to count annual layers between 5.3 m and 770.1 m depth. The deconvolution is sensitive both to melt layers and to
unusually wide layers. The first contain sharp gradients which create artefacts in the data, typically resulting in a series of high-amplitude oscillations that do not correspond to real annual layers, while the second result in spurious low-amplitude oscillations. The width of these perturbations is usually 2-5 years, and they are not difficult to spot for a trained investigator (Supplementary Information Fig. S7).

### 2.5. DYE-3

The oldest ice core in this chronology is DYE-3 whose drilling was completed in 1981 (Clausen et al., 1988). The data available to our study are mainly ECM and water isotopes (Langway et al., 1985). The isotope record resolves the annual layers very well thanks to the high accumulation rate which provides wide layers that are safe from diffusion, and is used for counting from 0.9 to 1271.7 m depth. The ECM signal also appears to have an annual pattern, hence we also used ECM to count layers from 136 m to 1271.7 m. Because of lacking setup in the first year of drilling, the ECM measurements only start after
136 m. Therefore, to construct the top chronology of DYE-3, we included two shallow cores named 4B and 18C, for which ECM and water isotopes were available for counting (Vinther et al., 2010).

## 3. Methods

The first objective of this study was the construction of a common chronology for several ice cores with data suitable for annual layer counting.

Our timescale construction method relies on three main steps:

- Automated annual-layer boundary identification using SC;
- Ice-core matching using volcanic and ammonium tie-points;
- Multi-core layer comparison by multiple observers (called fine-tuning).

A study of the uncertainty of the timescale completes our method.

### 3.1. The raw output: counting annual-layers on each ice core with StratiCounter


To avoid the lengthy process of manual layer counting, GICC21 was based on a multi-core set of annual layers identified on each ice core by SC, which also returns an uncertainty distribution of the layers of each individual ice core. SC has better performances with multiple proxies, but including more than four species did not prove to make a substantial difference for the final result, because some species are not independent of each other (e.g. those dominated by minerals
with dust as primary source) and some have similar seasonal patterns. As training data, SC requires a set of annual layers manually placed by the user. We chose to place the annual layer mark on the annual sodium maximum as the best indicator of the start of a new year, except for ice cores without impurity data, where we chose the isotope annual minimum, since the two methods are equivalent (Supplementary Information Fig. S8).

Measurement gaps should be minimized using all available data to obtain an accurate layer count. Since DEP is generally
measured on the full ice core, and ECM is measured on the first longitudinal cut of the core, they are both less affected by ice core breaks than the subsequent measurements made on smaller samples or obtained from continuous stream of melted sample. So, although the yearly pattern in the ECM signal is not always discernible and cannot be the basis of reliable annual layer identification, it proved useful for ice cores with many small gaps, like NEEM and EastGRIP. In addition, the ECM records of DYE-3 and, to some degree, GRIP do exhibit an annual ECM cycle, which helps improve the SC result. When
data gaps cannot be avoided, SC makes a probabilistic estimate of the layer count considering the neighboring data.

To facilitate the pattern recognition process by SC, the datasets were preprocessed using the appropriate settings for each ice core, found on control data between the Laki (1783 CE) and Samalas (1258 CE) eruptions. For all data sets, the mean is subtracted. For NEEM and EastGRIP, missing values are interpolated across data gaps, since this improved the performance of SC over short data gaps. Impurity records ($Na^{2+}$, $Ca^{2+}$, $NH_4^+$, BC, and $NO_3^-$) were log-transformed and then z-score-
transformed over windows with a width of twice the expected average layer thickness. Elemental and ionic concentrations were treated identically, as the differences should not matter for layer identification. A summary of the preprocessing steps is presented in the Supplementary Information. Even though it is possible to constrain SC to historical age markers, we chose to run SC in unconstrained mode to be able to quantify any possible biases of the algorithm.



In order to account for changes in layer thickness or data quality in each ice core, a variant of SC was implemented to count
on independent stretches of data. This requires that, upon a new run of SC, the probability priors of the precedent data
count are included for counting in the next adjacent section (more details can be found in the Supplementary Information
Table S1).

We observed that SC has a tendency to under-count over data gaps, especially within longer gaps. This issue was fixed *a
posteriori* by evaluating the average layer thickness around each gap and inserting the missing layers. However, too large
datagaps make the timescale unaccurate. Around 3.8 ka b2k, the length of data gaps in NEEM and EastGRIP increases as
both ice cores enter the brittle ice zone. Around the same time, the effects of isotopic diffusion in GRIP gradually make
recognition of the annual signal difficult, and the high-resolution sampling was discontinued (Vinther06). Therefore, we
stopped the timescale revision at 3835 years b2k in order to follow our multi-core data requirement. To continue further
back in time, the timescale would have to rely only on EastGRIP and DYE-3 data, which appear to be quite different and
therefore hard to synchronize; moreover, there are increasing EGRIP CFA data gaps until the end of the brittle zone. This
scenario calls for a different approach than the methods used here and will be dealt with in the future.

### 3.2. Ice-core matching using synchronous events

Ice core time scales were matched to each other by finding patterns of assumed synchronous events that will be referred
to as tie-points. Previously published ice-core matches (Rasmussen et al., 2013; Mojtabavi et al., 2020) were extended to
all cores considering the new data sets. The manual match is facilitated using a MATLAB GUI called Matchmaker that allows
for the insertion of visual bars to place stratigraphic markers on top of the data and to align the data according to these
markers (Rasmussen et al., 2008), of which an example is given in Figure 2.

Patterns of $NH_4^+$ spikes consolidate the volcanic match when eruptions are widely spaced by many decades. Because the
$NH_4^+$ patterns are mostly recognizable in ice cores from close sites (for example in NEEM and NEEM-2011-S1), we refer to
them as major or minor ammonium tie-points depending on how many ice cores they appear in. The frequency of the
ammonium spikes is on the order of one in 10 years, so that, by including ammonium, we have effectively increased the
resolution of the multi-core match with respect to a purely volcanic match. As mentioned before, the ECM shows some
narrow and deep valleys that strongly correlate with $NH_4^+$ spikes across ice-cores. Hence, for cores lacking $NH_4^+$ records, we
inverted and log-transformed the ECM record and used it as an ammonium-substitute, which we will call pseudo-$NH_4^+$ in
the following (reported on arbitrary units).

The timescale reference datum is the Laki eruption that happened between June 1783 and February 1784 CE, which is easily
detected in all ice cores thanks to a most pronounced acidity spike and to a corresponding tephra deposit (Clausen &
Hammer, 1988). For DYE-3, however, we tie the timescale to the Öræfajökull eruption of 1362 CE because the DYE-3 ECM
measurements start below the Laki layer.

When reporting historical events, we find it most convenient to use CE/BCE years. When talking about the ice-core
timescale, we will use years b2k, which are converted to CE by subtracting the b2k age from 2000 CE. Years BP, common
in the paleoclimate literature, can be translated by simply adding 50 years (i.e. 1000 BP = 1050 years b2k). Because there
is no year 0 CE, we subtract one year from the conversion whenever the b2k year is greater than 1999 (e.g.
2000 CE - 2000 years b2k -1 = -1 CE). Since years b2k increase going back in time, we remark that decimal ages correspond
to the inverted month order. For example, the exact layer boundary 200.0 b2k corresponds to January $1^{st}$ 1800 CE, while
200.1 b2k corresponds, approximately, to December $1^{st}$ 1799 CE, and 200.9 years b2k is February $1^{st}$ 1799 CE. We
mention that it is not necessarily possible to perform sub-annual dating of events, since accumulation throughout the
year is not constant. In the timescale data file provided with this paper, we give more details about the age calculation,
the years b2k convention, and the datum of this timescale.






Figure 2 An example of the matching of ice-core data. Data is shown between three eruptions, which are highlighted by grey, vertical bars: tie-point no. 1, no. 14 (Eldjá, 939 CE), and no. 15. Proxies of volcanic acidity are shown on each ice-core's top axis. Proxies for biomass burning events are shown on the bottom axes, including the pseudo-$NH_4^+$ obtained from inverting and log-transforming ECM. Turquoise bars highlight the patterns of $NH_4^+$ that were used to consolidate the match between the more widely spaced volcanic eruptions. No. 2, 3, 7, 13 show best that $NH_4^+$ spikes correlate with ECM dips (that is, pseudo-$NH_4^+$ peaks). We observe variable amplitude in volcanic signature, for example no. 15 has a different relative amplitude in each ice core.



### 3.3. Requirements for a manual fine-tuning procedure

The number of layers between synchronous tie-points must be the same across ice cores. Hence, we assessed the quality of the SC output by evaluating the number of layers counted by the algorithm between historically known volcanic
eruptions. Even after having chosen the best settings for SC, we observed a general tendency of under-counting in some ice cores and over-counting in others (Table 2). This is a result of the algorithm being run without age constraints, which was considered to be appropriate for using the method in older sections of data, where no age constraints are known. We conclude that, although SC correctly identifies almost all annual layers, it fails to assign a sufficiently high probability to some of the thin or otherwise unusual annual layers in each ice core, which are thereby not counted. Also,it assigns an
excessive probability to other layers, which upon closer inspection cannot be confirmed as annual layers, especially after comparison with the age of historically known eruptions.  The cause of the under-count may be related not only to data gaps but also to partially wind-eroded layers of snow, to unusual impurity loading, or to other site-specific perturbations. In the case of DYE-3, the high accumulation makes it unlikely to miss layers in the count but, in turn, gives higher risk of multiple isotope oscillations within one year, leaving more opportunities for an over-count of the layers, which could also
result from the higher occurence of melt layers in this ice core. Therefore, we resort to a manual processing of the timescale which in the following we will refer to as "fine-tuning".

We remark that fine-tuning is not in contradiction with SC, since the algorithm is, after all, trained to recognize the layers rather than to count them, meaning that the algorithm does not aim at speculating if the layer-count provides an accurate timescale or not. SC assigns a probability threshold to the data based on the layer pattern, hence we have the
right to influence the final count by adding and removing uncertain layer boundaries, a task we approach with the method of ice-core comparison.

Table 2 Evaluation of the SC bias of each ice core in the timescale sections constrained by historical evidence about the eruption ages. The eruptions chosen for the overview were all icelandic, except for the Indonesian eruption of Samalas : Laki (1783 CE), Hekla (1512 CE), Bárðarbunga (1477 CE), Oræfajökull (1367 CE), Samalas (1258 CE). Within each interval, the number of expected layer boundaries
is indicated by *N*. For each ice core, the difference between N and the number of layers counted by SC shows the amount and the direction of the bias. A positive value means that layers had to be added in order to reach the expected N, while a negative value means that they had to be removed. The longest interval (Laki-Samalas) shows the total layer-modifications required by each ice core. The last column sums all ice cores to indicate the overall tendency of the fine tuning: a notable amount of additions is required to correct for the bias in individual SC counts. The measurements indicate that the "CFA-cores" (EastGRIP, NEEM, NEEM-2011-S1) are
mostly under-counted. On the other hand, the "GICC05-cores" (NorthGRIP1, GRIP, DYE-3) were mostly in need of removing layers. This evaluation of bias is what eventually justifies the fine-tuning process over a purely statistical combination of SC results.

| Event | Age [b2k] | Event | Age [b2k] | N | EastGRIP | NEEM | NorthGRIP1 | NEEM-2011-S1 | GRIP | DYE-3 | TOTAL |
|---|---|---|---|---|---|---|---|---|---|---|---|
| Laki | 216.5 | Hekla | 487.4 | 271 | 7 | 5 | -4 | -1 | -1 | -4 | 2 |
| Hekla | 487.4 | Bárða. | 522.8 | 35 | 1 | 2 | 0 | 0 | 0 | 1 | 4 |
| Bárða. | 522.8 | Oræf. | 637.3 | 115 | 0 | 3 | -1 | 3 | 0 | 0 | 5 |
| Oræf. | 637.3 | Samal. | 741.1 | 104 | 2 | 0 | 2 | 0 | 1 | -2 | 3 |
| Laki | 216.5 | Samal. | 741.1 | 525 | 10 | 10 | -3 | 2 | 0 | -5 | 14 |
| | | Of which layers added in gaps | | | 2 | 5 | 0 | 2 | 0 | 0 | 9 |

The SC raw layers are accompanied by a probability distribution which represents the likelihood of the placement of each single layer. The fine-tuning was guided by observing where the likelihood of the layer placement is most unsure, since the
SC uncertainty increases locally where the data-quality is low. Here, the 95%-percentiles of the probability distribution register a 1-year "jump" that can be used to detect the layers that SC deemed to be most uncertain. On the other hand, we also measured the placement of "ghost layers" where the probability was just below the threshold at which SC assigns an annual layer.

The fine-tuning is performed by comparing all ice cores in parallel and using an iterative protocol. To ensure reproducibility,
we adopted a ruleset to the fine-tuning process: we added layers in gaps according to the local layer thickness; we removed low-probability layers that conflicted between parallel cores; we upgraded ghost layers to full annual layers when indicated by parallel-core comparison. Some examples of how the fine-tuning was done can be found in Supplementary Information Fig. S 1.

Minor similarities between the records, such as minor ECM or $NH_4^+$ features, $\delta^{18}O$ patterns, and in some cases similar peak-
shape sequences in $Na^{2+}$ or $Ca^{2+}$ for geographically close ice cores, were used to support or reject changes in the layer count. As a consequence, some tie-points had to be re-examined because of now-apparent misalignments, and the fine-tuning was repeated to ensure consistency between the ice cores.



As a further step, two to three observers were engaged in detailed review of each section of the timescale aiming to reduce the impact of potential confirmation bias by each investigator. Whenever unanimity was lacking, the main observer (Sinnl) examined the different opinions to propose a final solution, which was then accepted or rejected again. In the end, unanimity was reached in all sections.

### 3.4. Top-chronology and remarks about the DYE-3 ice core

The DYE-3 ice core is the best ice core to directly compare our results to GICC05, because its well-preserved isotope signal provided the foundation for GICC05 in the last 4k years. Hence it is important to match it precisely, even for the top 136 m where the ECM measurements are missing. To match the top part of DYE-3, we used two shallow cores, 4B and 18C, drilled respectively 8 km and 36 km upstream of the main drilling site for climate reconstructions (Vinther et al., 2010). Thanks to the close site of the three ice cores, their $\delta^{18}O$ records could be matched by finding common features in the data (Figure 3). The ECM signal of the two shallow cores was in turn matched to NorthGRIP1 and GRIP. In this way, DYE-3 has been tied to the timescale for the top 136 m.

Vinther et al. (2010) showed that Greenland ice-core isotope signals correlate well with South West Greenland temperature records, and that the inter-core correlations (e.g. between DYE-3 and GRIP) help assess the quality of the cross-dating of the ice cores. To judge the quality of our own top chronology, we repeated the correlation study of DYE-3 with the South West Greenland (SWG) temperature and with GRIP (Table 3). The correlation between isotopes of the DYE-3 main core and GRIP improved by 17%, a promising result that indicates that the matching between DYE-3 and GRIP has improved. However, the correlation with SWG temperatures is essentially the same as in Vinther et al. (2010).

Fine tuning of DYE-3 was made more difficult by the ice core's very different characteristics. Upstream surface undulations (Reeh et al., 1985) lead to low-frequency layer thickness fluctuations at depth (Figure 7). Many layers of DYE-3 show a double isotopic oscillation during the year, a fact made clear during the top match with the shallow cores 4B and 18C (Figure 3). As shown in Table 2, the DYE-3 ice core was most affected by over-counting, most likely because of mid-year local minima. We speculate that mid-year oscillations in the isotopes of DYE-3 are related to melt layers and to the windy conditions that perhaps cause a redistribution of the snow in the area. In cases of doubt about where to place the annual layers, we decided to keep the layer boundaries of the GICC05 timescale where possible without creating annual-layer thickness outliers. Finally, DYE-3 proved to be the most difficult to match, its ECM record being noisy and not strongly resembling that of the other cores, since it also forms spikes because of the influence of high $HNO_3$ levels (Clausen et al., 1997).

Table 3 Correlations (ρ) between $\delta^{18}O$ in the DYE-3 ice cores and the SWG temperatures or the GRIP $\delta^{18}O$ over the last 400 years. The correlation between GRIP and all the DYE-3 ice cores (main core 79 and shallow cores 4B and 18C) have increased substantially with the new GICC21 match. No significant change is observed in the correlation between DYE-3 main core (DYE-3 79) and SWG temperatures, and no significant change is observed in the correlation between the shallow cores and SWG temperatures.

| | DYE-3 79 | | DYE-3 4B | | DYE-3 18C | | GRIP | |
| --- | --- | --- | --- | --- | --- | --- | --- | --- |
| | | | Vinther et al., 2010 | | Vinther et al., 2010 | | | |
| | GICC05 | GICC21 | | GICC21 | | GICC21 | GICC05 | GICC21 |
| ρ SWG | 0.46 | 0.47 | 0.45 | 0.44 | 0.35 | 0.36 | 0.21 | 0.20 |
| ρ GRIP | 0.23 | 0.27 | 0.28 | 0.31 | 0.24 | 0.26 | | |



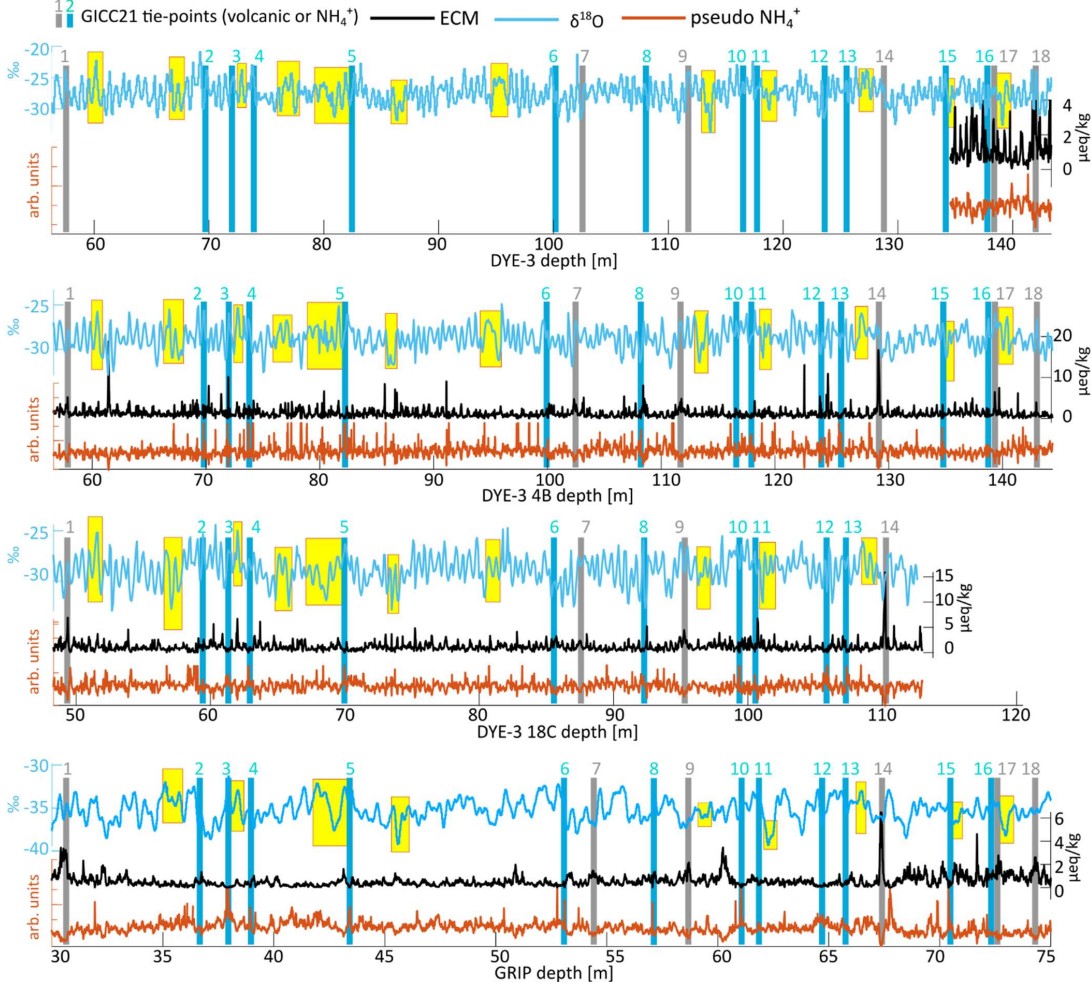

Figure 3 Matching the top of DYE-3 using the two shallow cores 4B and 18C. Tie-point n.14 is the Laki eruption (1783 CE). Data for 18C covers only until the Laki eruption, while 4B offers a longer overlap with the start of the ECM signal in DYE-3. Yellow patches highlight some of the common features in the isotopic records that were used to match DYE-3 in absence of ECM measurements. Some of the matched features where also visible in GRIP (although GRIP was not used directly for the isotope-based matching). We observed mid-year oscillations in the DYE-3 isotopic signal, visible for example between tie-points no. 13 and 14, a knowledge we applied to interpret DYE-3 layers further on in the chronology.

### 3.5. Uncertainty of the GICC21 chronology

An important part of our objectives is to provide a simple yet empirically justified estimation of the uncertainty that is associated with the GICC21 timescale. Uncertainties in the layer count arise from two main sources: issues with the data and misinterpretation of layers (Vinther06). In our study, data gaps are a prevailing issue because the ends of each ice-core piece are trimmed to prevent contamination during CFA measurements. This, combined with removal of small pieces around core-breaks from the drilling process, causes frequent but brief interruptions in the records. We expect the misinterpretation of layers to be largely accounted for by the fine-tuning of multiple parallel ice-core records. For the part of the timescale younger than the Samalas eruption (1258 CE), the uncertainty is constrained using historical evidence. For the deeper part of the chronology, without any independent time constraints to rely on, issues with layer identification accumulate, causing the uncertainty envelope to expand with time.





There are many factors that increase the complexity of the uncertainty estimation for our new ice core timescale. We find ourselves in a mixed scenario between automated counting by SC, an algorithm that provides its own probability estimates, and our manual intervention by fine-tuning. Moreover, errors are not completely uncorrelated, because of local effects within the layer count, a fact that calls for an investigation of the typical correlation length. An example of such an effect is the possible local bias given by visualizing a large amount of ice core data at once, which is likely to be independent from the data sections outside the visualized window. We are not aware of any theoretical basis that adequately describes such 465 a scenario. Hence, we have addressed this question by a semi-empirical approach, which considers data issues and possible counting biases.

Until the Samalas eruption of 1258 CE (742 years b2k), the fine-tuning is constrained by well-established historical evidence on volcanic eruptions. Because of transport and deposition dynamics, the volcanic signals in the ice cores are affected by delays, which we quantify to be within 1 year after the event started. Furthermore, we estimate that an additional 1-year 470 contribution to the uncertainty originates from possible inaccuracies in the annual layer positioning between the eruptions. Therefore, we argue that the uncertainty is never below 2 years, but also that this is a realistic total uncertainty of the historically constrained part of the timescale.

For the remainder of the timescale, we proceed by quantifying how the changes made during multi-core fine-tuning compare to the uncertainties provided by SC for each ice core. The probability distributions of the ice cores, as provided by 475 SC, can be averaged by convolution. The convolution's width, which can be depth/age-dependent because of data quality and coverage, would be a suitable candidate for the uncertainty in the absence of bias. However, as we have argued above, there is a bias induced by SC that needs to be accounted for. The quantification of the SC-induced bias in Table 2 cannot be extended to the older sections of the timescale, since we do not know *a priori* how much the true count would be. Hence, we can only compare the SC-output to the already fine-tuned timescale to gather some empirical evidence of the SC-bias.

In order to assess the SC-induced bias for ages older than 1258 CE, we conducted a comparison of the SC output and the fine-tuned results in sections of 100, 200, and 300 fine-tuned years, at regular intervals covering the study period. In each section, we performed a SC run for every ice core, acquiring the independent probability distributions of the annual layers (prob-SC), for which plots can be found in Supplementary Information Fig. S3. We manually accounted for the under-count across data gaps based on the local layer thickness (Supplementary Information Table S2). The individual prob-SC were 485 convoluted (conv-SC) to provide the multi-core average SC-estimation of the number of layers in the ice. A Gaussian curve was fitted to the conv-SC to obtain a mean and a standard deviation, indicated by conv-SC-$\mu$ and conv-SC-$\sigma$ (Table 4).

Table 4 Means and standard deviations of the Gaussians fitted to the conv-SC. The absolute difference of the conv-SC-$\mu$ from the fine-tuned value (i.e. the window length) is weight-averaged to quantify the SC-induced bias.

| | | | Mean: conv-SC-$\mu$ (years) | | | | | | Standard deviation: conv-SC-$\sigma$ (years) | | |
|---|---|---|---|---|---|---|---|---|---|---|---|
| | | Start (year b2k) | Window length (years) | | | | | Start (year b2k) | Window length (years) | | |
| Section | | | 100 | 200 | 300 | | Section | | 100 | 200 | 300 |
| 1 | a | 742 | 99.46 | 198.32 | 298.26 | | 1 | a | 742 | 1.77 | 2.17 | 2.31 |
| | b | 842 | 98.40 | 198.40 | | | | b | 842 | 1.31 | 2.09 | |
| | c | 942 | 99.80 | | | | | c | 942 | 1.91 | | |
| 2 | a | 1700 | 99.55 | 198.19 | 300.71 | | 2 | a | 1700 | 1.03 | 1.86 | 1.85 |
| | b | 1800 | 98.84 | 200.39 | | | | b | 1800 | 1.58 | 1.62 | |
| | c | 1900 | 100.12 | | | | | c | 1900 | 1.16 | | |
| 3 | a | 2700 | 99.61 | 200.63 | 300.68 | | 3 | a | 2700 | 0.96 | 2.01 | 2.16 |
| | b | 2800 | 99.63 | 197.91 | | | | b | 2800 | 1.69 | 3.48 | |
| | c | 2900 | 96.87 | | | | | c | 2900 | 2.40 | | |
| 4 | a | 3500 | 98.44 | 201.17 | 303.66 | | 4 | a | 3500 | 1.74 | 2.20 | 2.41 |
| | b | 3600 | 99.58 | 202.04 | | | | b | 3600 | 1.90 | 2.71 | |
| | c | 3700 | 101.56 | | | | | c | 3700 | 2.00 | | |
| Average absolute difference ($\mu$-100) | | | 0.95 ± 0.20 years | | | | | | | | | |

As expected, the single ice cores most often do not reach the fine-tuned value, i.e. the conv-SC-$\mu$ tends to be less than the fine-tuned window length. Moreover, since we do not observe a depth-dependence in these results, we assume that the results are  transferrable to the remaining part of the timescale.  The average absolute difference of the conv-SC-$\mu$ from



the fine-tuned value is 0.95 ± 0.20 (1 σ) years per century and quantifies the bias of SC in our work, meaning that, on average, about one layer per century is mis-interpreted by SC in all the cores, although it is not necessarily the same layer.

Having acknowledged the existence of the SC-induced bias for the entire timescale, the already obtained prob-SC were fine-tuned by adding the layers that remained to be accounted for (besides to the already performed gap undercount correction). We remark that, since the SC counts in the test sections are produced by the same preprocessing and preliminary settings as the general timescale, one can expect the same layers to be missing or exceeding, even when we have run SC on shorter sections of data. Hence, we added and removed the same layers we found during the fine-tuning

process.

To test the leftover uncertainty purely related to the SC distributions, we computed a product distribution (prod-SC) of the fine-tuned prob-SC, assuming that the fine-tuned single-core layer counts are independent estimates of the same property, i.e. the true number of years in the particular section (Figure 4). Not all product-distributions are perfectly centered around the fine-tuned value, because of small asymmetries between the single distributions (Table 5). The SC-

related uncertainty averages at 0.24 years per century. The fine-tuning process leads to an added uncertainty, which is difficult to quantify independently, but very likely smaller than the bias correction itself, estimated above to 0.95 years per century on average. Based on the values obtained from the test sections, we suggest a depth-independent empirical uncertainty of 1 year every 100 layers.

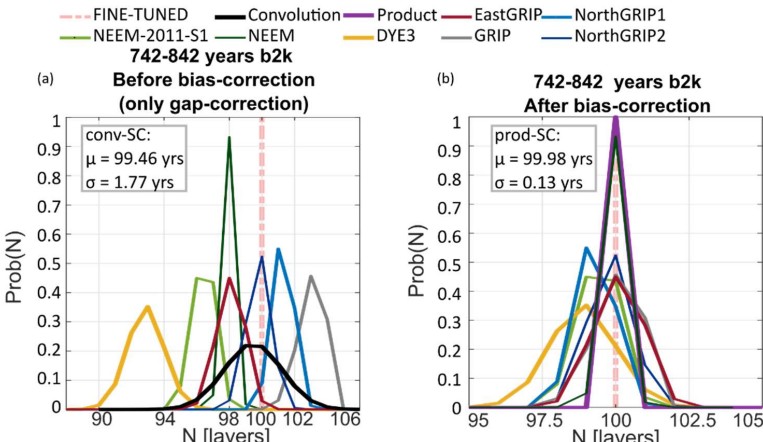


Figure 4 Probability distributions in the 100-year window between 742 and 842 years b2k (1258-1158 CE). (a) By only correcting the gaps, SC still shows some issues with the pattern recognition, which we call SC-induced bias. The conv-SC-μ is below the expected value of 100, and the conv-SC-σ is larger than 1 year. (b) The fine-tuning results in a much better centering of all individual prob-SC. The joint-probability distribution is given by the product of the fine-tuned prob-SC: it is centered around 100 and has a much narrower standard deviation.





Table 5 Means and standard deviations of the Gaussians fitted to the product distributions for the investigated time windows. The average standard deviation estimates the uncertainty we would have if SC was bias-free.

| | | Mean: prod-SC-μ (years) | | | | | | Width: prod-SC-σ (years) | | | |
|---|---|---|---|---|---|---|---|---|---|---|---|
| | | Start (years b2k) | Window length (years) | | | | | Start (years b2k) | Window length (years) | | |
| Section | | | 100 | 200 | 300 | Section | | | 100 | 200 | 300 |
| 1 | a | 742 | 99.98 | 200.37 | 300.59 | 1 | a | 742 | 0.13 | 0.29 | 0.31 |
| | b | 842 | 100.68 | 200.54 | | | b | 842 | 0.20 | 0.27 | |
| | c | 942 | 100.04 | | | | c | 942 | 0.28 | | |
| 2 | a | 1700 | 100.06 | 200.58 | 301.35 | 2 | a | 1700 | 0.12 | 0.28 | 0.39 |
| | b | 1800 | 100.95 | 200.42 | | | b | 1800 | 0.11 | 0.34 | |
| | c | 1900 | 100.78 | | | | c | 1900 | 0.26 | | |
| 3 | a | 2700 | 100.88 | 200.01 | 300.49 | 3 | a | 2700 | 0.31 | 0.33 | 0.50 |
| | b | 2800 | 100.02 | 201.91 | | | b | 2800 | 0.08 | 0.60 | |
| | c | 2900 | 100.38 | | | | c | 2900 | 0.43 | | |
| 4 | a | 3500 | 99.66 | 201.00 | 300.25 | 4 | a | 3500 | 0.34 | 0.07 | 0.36 |
| | b | 3600 | 99.82 | 201.94 | | | b | 3600 | 0.26 | 0.48 | |
| | c | 3700 | 101.41 | | | | c | 3700 | 0.34 | | |
| | | | | | | Average σ | | | 0.24 ± 0.11 years | | |

Now, we are left with the question of how to combine the uncertainty of adjacent centuries in a meaningful way. We observed that a volcanic tie-point is generally found at least once every 100 years. Because a layer identification error is
not likely to affect the section on the other side of a volcanic tie point, we find it reasonable to assume that, beyond 100 layers, errors in the fine-tuning are uncorrelated. Conversely, because the number of years between tie-points must match, errors in layer identification are likely correlated over shorter intervals.

Using this assumption, we apply the properties of normal statistics by investigating if the standard deviations of our joint-distributions (prod-SC-σ of Table 5) sum up in quadrature for adjacent centuries, that is if:

$$\text{prod-SC-}\sigma^2_{1st\ century} + \text{prod-SC-}\sigma^2_{2nd\ century} = \text{prod-SC-}\sigma^2_{both\ centuries}$$

and similarly, for the 2nd and 3rd adjacent centuries in our test periods. We present the results for all test periods in Figure 5 and conclude that this relation is justified.

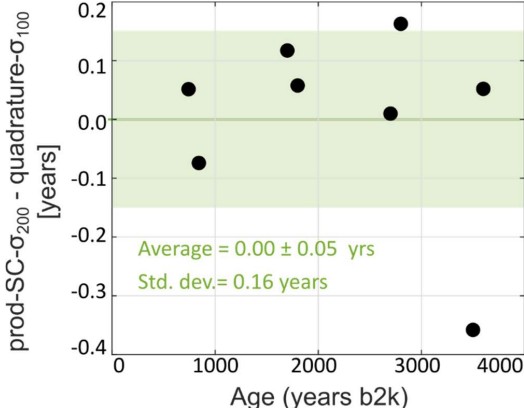

Figure 5. Study of correlation distance within the timescale. The prod-SC-σ for 200-year intervals are compared to the quadrature sums
of adjacent prod-SC-σ for 100-year intervals. The difference (black dots) is compatible with 0 years, showing that it is possible to use normal statistics to combine uncertainties across century-long windows.





The result supports our assumption that uncertainties of neighboring centuries can be treated as uncorrelated. Hence, we postulate that in the section older than Samalas, we can add 1 year in quadrature for every century of the timescale.

To conclude, we propose an uncertainty formula that incorporates our empirical observations, where the uncertainty between Samalas (age symbol: $S$ = 1258 CE = 742 years b2k) and any older age $t$ will have an absolute uncertainty of:

$$\delta t(t\,;\,t > S) = 2 + \sqrt{\frac{t - S}{100}} \quad \text{Years} \tag{1}$$

### 4. Results: the timescale offset curve

We now present a comparison between the new GICC21 timescale and the existing ice-core chronologies GICC05 (Vinther06) and NS1-2011 (Sigl15), investigating any dating-offsets (Figure 6). To calculate the GICC21 ages at reported GICC05 depths and infer the correct offset, we linearly interpolated the GICC21-ages on the GICC05-layers of DYE-3, GRIP, and NorthGRIP1 (Vinther06). A volcanic match of EastGRIP to NorthGRIP (Mojtabavi et al., 2020), and of NEEM to NorthGRIP (Rasmussen et al., 2013), allows us to find the GICC21-ages for these two cores at the published tie-points.

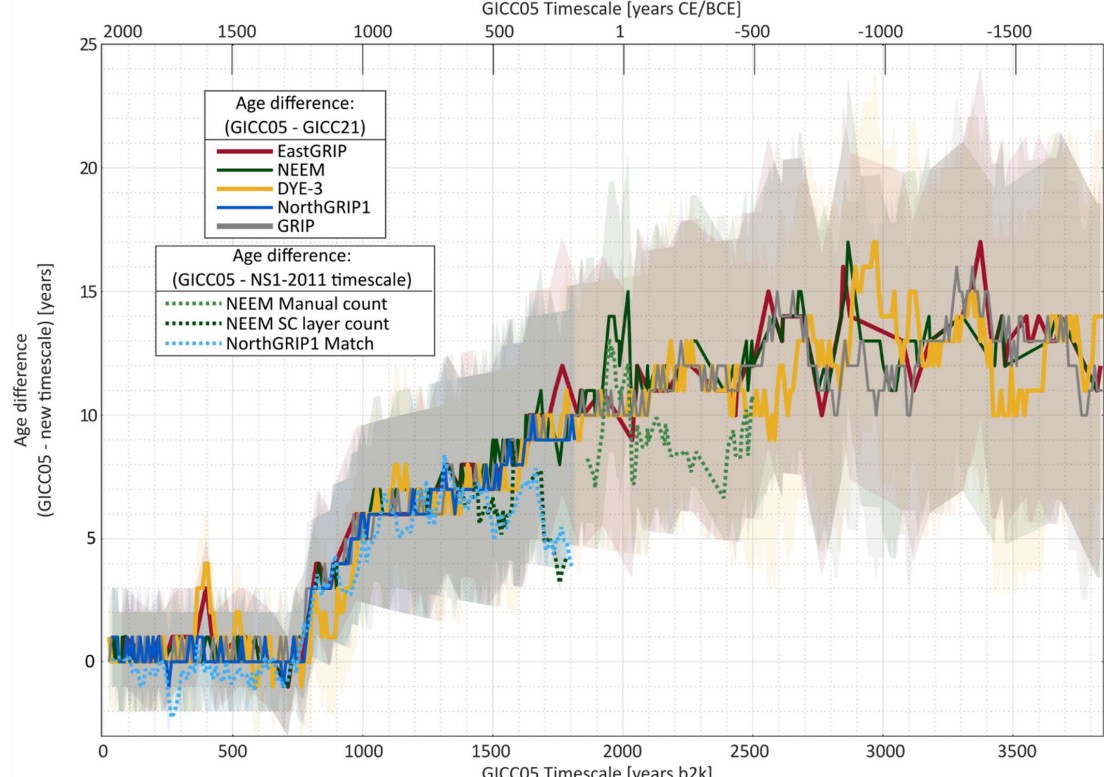

Figure 6 Timescale comparison for each ice core. A direct comparison with GICC05 is possible for the ice cores NorthGRIP, GRIP, and DYE-3, for which both GICC05 and GICC21 are annual-layer-counted (Andersen K. et al., 2006). The NorthGRIP layer comparison stops at 1813 years b2k, corresponding to the end of the IC dataset, after which GICC05 was transferred to NorthGRIP from the other two ice cores. An indirect comparison was possible for EastGRIP and NEEM, for which published match-point ages were used to interpolate to the new layer-counted ages (Rasmussen et al., 2013; Mojtabavi et al., 2020). As a further reference, the NS1-2011 timescale, which was built by layer-counting in NEEM and NEEM-2011-S1 records, is also compared to GICC05 (dotted lines). All of the ice cores show an increasing time offset, which is very steep between 750 and 1000 years b2k and building up to 10 years around the depth of the horizon formerly attributed to the Vesuvius eruption. The uncertainty for each curve is calculated using equation (1).





In Figure 6, all curves tend to stay close to each other, making it possible to identify a common behavior, which illustrates the result of the timescale revision. A drift of one ice core away from the others can mean two things: either the volcanic match is different, or there have been interpolation problems in GICC05, e.g. in the case of NorthGRIP1 beyond 1813 years
b2k. No offsets are observed in the section younger than about 750 years b2k, right after the prominent Samalas eruption of 742 years b2k, except around 390-490 years b2k, where DYE-3 displays a layer thickness fluctuation, discussed in Figure 7. Because EastGRIP is also matched differently, a divergence also arises there.

The erroneous attribution of Hekla (1104 CE) explains the steep 4-year offset which was introduced between the Samalas eruption and 1104 CE. In hindsight, the 4 annual layers appear poorly supported by the DYE-3 isotope data (Figure 8). After
1000 years b2k, a steady increase in offset is observed until about 2000 years b2k at the place of the previously assigned Vesuvius layer.

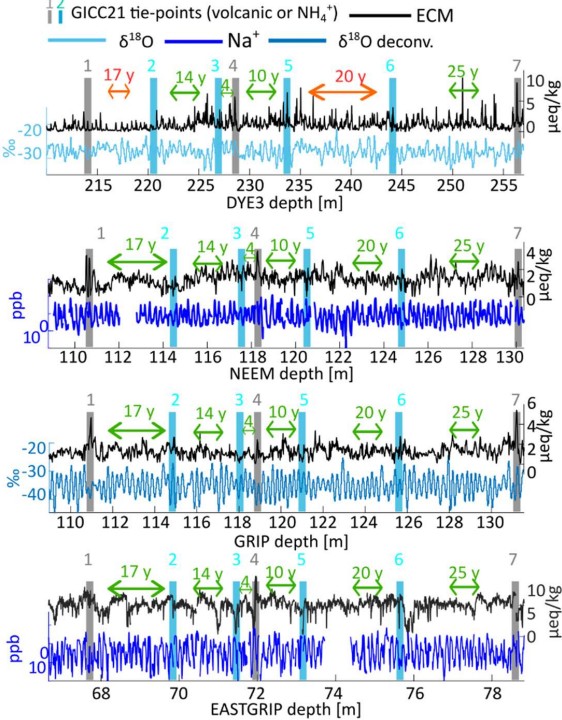

Figure 7 A closer look at the first divergence of the timescale offset, at 390-490 years b2k (Figure 6), reveals a layer thickness fluctuation
in DYE-3. Volcanic tie-point no.4 in DYE-3 shows a moderate displacement to the left, while the other ice cores have linearly spaced tie-points. The correctness of match points 1 (398 years b2k) and 7 (488 years b2k) is strongly confirmed by the matching of adjacent sections (not shown). The arrows highlight how many layers are found within tie-points. The section of relatively thin layers on the left and the relatively wide layers on the right of match point 4, forms this layer thickness fluctuation that spans about 40 m of ice in DYE-3. For EastGRIP, we observed that the broad shape of no.1 caused the match to be revised and placed 2 years later than was done by Mojtabavi
et al. (2020).



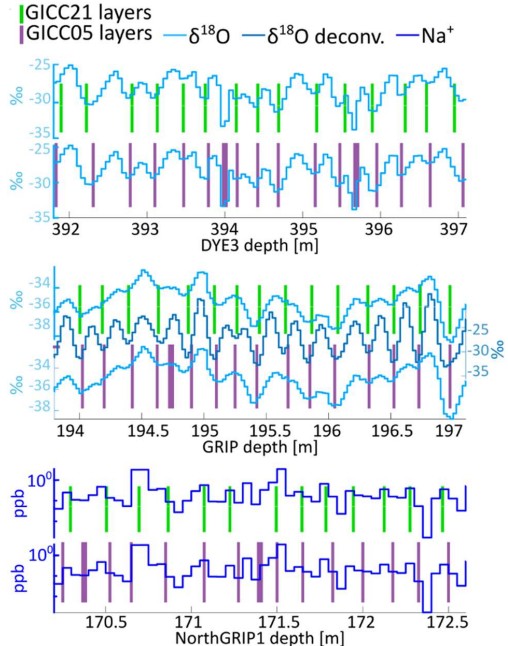

Figure 8 Example of GICC05 overcounting in sections of DYE-3, GRIP, and NorthGRIP1, in a short window between Samalas and 1108 CE. The top, green bars represent the GICC21 layer boundaries. The bottom, purple bars represent GICC05 layers: they include more layers (thicker purple bars) that are not included in GICC21. For DYE-3, the doubtful features of the isotopes, possibly corresponding to melt layers or measurement issues, suggest that these are not annual layer boundaries. For GRIP, it seems unlikely that a year is found at a local maximum of the isotopes. For NorthGRIP, discrete sodium measurements are not always easy to interpret due to the marginal resolution, however the placement of a layer boundary at a sodium minimum is unlikely.

### 4.1.1. The offset behavior between 2000 and 3835 years b2k

Beyond 2000 years b2k, the offset stays above 10 years, reaching an average of around 13 years in the last centuries. We speculate that the reason for the overall increase in offset is related to a confirmation bias in GICC05 after having acquired the initial 10 years offset, meaning that by deciding to include more layers before Vesuvius, the authors possibly continued to lean towards interpreting melt layers or isotopic fluctuations in DYE-3 as annual layers.

Around the previously attributed Vesuvius match, the NEEM ice core exhibits a divergence from the other ice cores, which was also documented in Sigl15. NEEM was matched to NorthGRIP1, which was matched to DYE-3 and GRIP, as the section sits just below where the NorthGRIP1 IC-data stops. We thus conclude that the likely reason of the fluctuation is a previously erroneous transfer of GICC05, because the problem must lie in the ages previously assigned to the NorthGRIP1 match points. Since EastGRIP was matched with fewer match points in this section, the divergence does not arise in the EastGRIP curve.

Between 2500 years b2k and 3500 years b2k, three centennial-scale fluctuations are observed, with two notable offset-peaks above 15 years at around 2900 and 3400 years b2k. We argue that these large wiggles in the timescale offset are to be attributed to a different layer count within widely separated volcanic markers. The spacing of volcanic eruptions can be as high as 130 years, a fact that called for a heavier use of $NH_4^+$ markers in our work, and which were not used in GICC05. For the timespan 2800-3100 years b2k, we analyzed in detail the matching differences between GICC05 and GICC21 (Supplementary Information Fig. S 4) finding that the offset wiggle is explained by shifts of the tie-points, by layer thickness fluctuations of DYE-3, and by interpolated NorthGRIP1-ages being used to date EastGRIP and NEEM.

### 4.1.2. The comparison with NS1-2011

By making a similar interpolation of the NS1-2011 timescale, we found that both GICC21 and NS1-2011 have a similar offset to GICC05 (Figure 6). A notably lower offset is observed at 1700 years b2k, most likely caused by the single-core approach of the NS1-2011 timescale. The NEEM manual count, on the other hand, is again getting closer to GICC21 at 1900 years b2k, producing a considerable jump in offset between 1700 and 1900 years b2k. There is also a lack of continuity in the curves since the NorthGRIP revised ages terminate at 1814 years b2k and the NEEM count is matched to





NorthGRIP again only after 1912 years b2k. NS1-2011 also shows an offset wiggle between 2000 and 2500 years b2k, well within the GICC21 uncertainties, that nonetheless might be a sign of widely spaced eruptions and lack of multi-core comparison.

## 5. Discussion

### 5.1. Estimating an offset between GICC21 and IntCal20

After having examined the offset between GICC05 and GICC21, we computed a transfer function that averages the separate contribution of the ice cores to the offset. The transfer function can be used to translate any age previously matched to GICC05 to the new revised GICC21 ages (provided in Timescale Supplement, see Appendix A for more details
on the transfer curve).

The ice-core timescale can be compared to other timescales and climatic archives to verify their consistency and infer leads and lags in the climatic system. GICC05 was compared to the radiocarbon calibration curve IntCal13 by Adolphi & Muscheler (2016), who found that the offset between the timescales increased steadily over the Holocene, reaching about 20 years at 4 ka b2k. This conclusion is supported by our new time scale at least until 3400 years b2k, by observing that the transfer
function of GICC05 to GICC21 behaves similarly to the one produced by Adolphi & Muscheler (Figure 9a), which indicates that the revised timescale is now in a better alignment with the IntCal time scale, i.e. ages from radiocarbon dated samples. Furthermore, the solar proton event identified by O'Hare et al. (2019) in NorthGRIP and GRIP keeps its alignment under the GICC21 timescale and is confirmed at an age of about 2610 years BP, providing independent proof at this age of the accuracy of both transfer functions for that period.

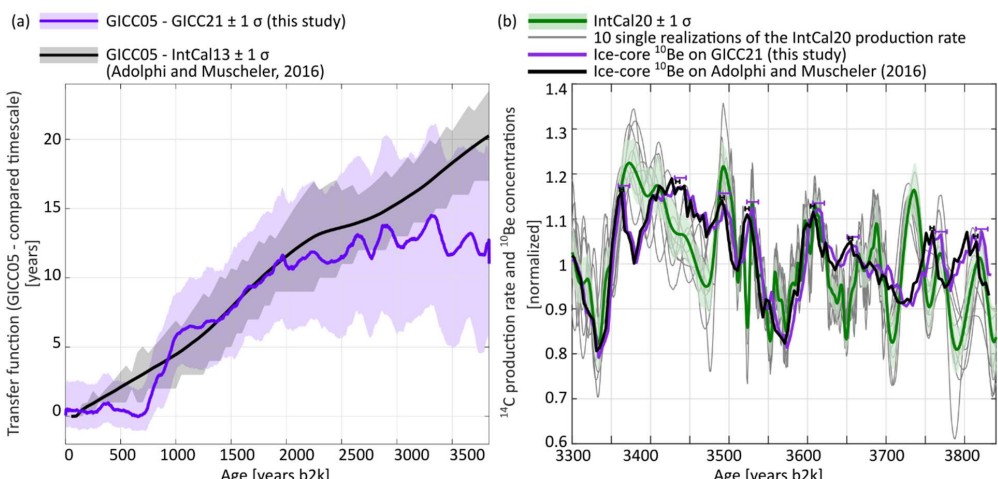


Figure 9 (a) Comparison of the transfer function GICC21-GICC05 with the transfer function modelled by Adolphi & Muscheler (2016), who used $^{10}$Be data from the GRIP ice core, converted it to $\Delta^{14}$C by modelling, and compared it to the $\Delta^{14}$C variations in IntCal13. The agreement between GICC21 and IntCal13 is supported by the closeness of the two transfer curves. We observe two notable differences between the transfer curves: between 500 and 1000 years b2k, where the effect of the 1000-year smoothing of the Adolphi & Muscheler
approach (used in the $^{10}$Be-$^{14}$C comparison to avoid matching spurious peaks) is evident, and possibly between 3400 and 3835 years b2k, where some offset (7±6 years) towards older GICC21-ages is still observed. (b) $^{10}$Be concentrations measured in the GRIP ice core (Muscheler et al., 2009) are shown with 20 years running average and detrended to remove the long term trend. The $^{14}$C production rate based on the tree-ring timescale is obtained from carbon-cycle modelling of the $^{14}$C data of IntCal20 (Muscheler et al., 2005). Single realizations of the IntCal20-based production curve show that the position of the peaks underlying IntCal20 can vary slightly, so that the
average curve should be handled with care when performing timescale studies (Muscheler et al., 2020). The ice-core data is either aligned to GICC21 (purple) or to IntCal13 (black), with horizontal bars highlighting the uncertainties in the peak positioning based on the transfer-curves' 1-σ. The GICC21-aligned data shows slightly older ages than the intCal13-aligned data. The alignment with IntCal20 is kept within uncertainties throughout the period shown, with possibly GICC21 better aligned in the last century. Any production rate differences between tree-ring data and ice-cores that cannot be resolved within realistic dating offsets (e.g. the period 3700-3740 years b2k) could
be explained by under-estimated data uncertainties or by transport and deposition effects on $^{10}$Be, since major carbon-cycle changes in this period are unlikely (Muscheler et al., 2004).

The transfer curve to IntCal13 is smoothed as a result of the statistical wiggle-matching approach between $^{10}$Be data and $^{14}$C data, designed to match unstretched 1000-year long windows in order to avoid over-fitting of spurious peaks (Adolphi



& Muscheler, 2016). This implies that beyond 3500 years b2k, the wiggle-matching algorithm is influenced by data older than 4000 years b2k, which could cause the higher observed offset in the 3500-3800 years window. Therefore, after 3500 years b2k, the IntCal13-GICC21 offset can be quantified as 7±6 years, i.e. not completely negligible.

To address the finer structure of the offset in the last 500 years of the GICC21 revision, we compare the ice-core $^{10}$Be concentration measured in the GRIP ice core (Muscheler et al., 2009) and the $^{14}$C production signal of the IntCal20 curve, which is obtained by carbon-cycle modelling (Muscheler et al., 2005). Since the underlying production mechanisms are the same, the two radionuclides show common variability. Furthermore, the offset to IntCal20 (Muscheler et al., 2020) is thought to behave in the same manner as for IntCal13, considering that the two calibration curves share the same tree-ring timescale in the Holocene, which maintains the positioning of signal peaks at the previous ages. After detrending the signals, we compare the production curves, and we furthermore align the ice-core data according either to GICC21 or to the transfer function to IntCal13. From Figure 9b we conclude that there is good agreement of the production signals within the dating uncertainties and that until 3835 years b2k the offset between IntCal20 and GICC21 is resolved within uncertainties. However, we remark that between 3700 and 3800 years b2k comparative studies of tree-ring and ice core data should address the inconsistencies observed in the radionuclide production signal. In conclusion, there is no compelling evidence to suggest an offset of GICC21 versus IntCal20.

### 5.2. The behavior of acidity, isotopes, and layer thickness around eruptions

A feature of the GICC21 multi-core timescale is the possibility to average the signals of all involved ice cores with annual resolution. We performed averages of ECM, $\delta^{18}$O, and layer thickness. We report more details on the layer thickness in Supplementary Information Fig. S 2, and on the stacking method in Appendix B.

After having obtained the signal stacks, we investigated the behavior of Greenlandic ice cores around volcanic eruptions. To do so, we averaged in sections around the 105 eruptions that were identified in this study as volcanic tie points, without 660 any distinction of origin. In this way, we obtained a detailed image of the average response recorded on the Greenland ice sheet following all eruptions. In Figure 10, we present the averaged stacks of ECM, $\delta^{18}$O, and layer thickness for 50-year windows around the eruptions. We observe a drop both in isotopes and layer thickness, which lasts for about a decade after the eruption. To assess whether this result is significant we performed 10'000 Monte Carlo iterations in which we sampled data from 105 random ages (details of analysis can be found in Appendix C). In each simulation, the amplitude was 665 calculated as the difference of the mean value before and after the eruption. Also, the duration of the minimum was calculated as the duration of the longest cluster of data below the mean.

The simulated results form a distribution that is Gaussian for the amplitude and almost-Poisson for the duration, both in the case of the $\delta^{18}$O data and of the layer thickness. For the latter, we took the precaution of excluding the top 100-years of the chronology from the simulation, because this caused the distribution to have excessive samples in the negative tail 670 of the Gaussian – a residual effect of firn densification. The measured amplitude jumps of $\delta^{18}$O and layer thickness lie, respectively, at the 99% and the 96% percentiles of the empirical distributions, which we find to be a noteworthy result. The duration of dips in isotopes and the accumulation have percentile values of 97% and 82%, respectively. The measured duration of 5 years for the layer thickness depends on the sensitivity of the algorithm used for the measurement, but visual inspection shows that 10 years is a valid estimate, especially considering a marked dip seen 10 years after the eruption. All 675 findings show the rare occurrence of such a long and deep signal-drop compared to the random sample.

The same study was repeated by using GICC05-data but did not return the same answers. On one hand, less data and a poorer alignment of the ice cores don't show any important minimum in the data (Supplementary Information Fig. S 4). On the other, using data from NorthGRIP, GRIP, and DYE-3, but aligned on GICC21, returned a minimum in isotopes, but not in layer thickness, that is less evident than what we find with the complete dataset. We conclude that the extensive dataset 680 and a better alignment are both necessary for reproducing this finding.

In summary, after a volcanic eruption the $\delta^{18}$O decrease on average by 0.12 ± 0.02 ‰, the layer thickness by 1.1 ± 0.3 % (normalized units, see stacking method in Appendix B), and both display a 10-year-long minimum.

A drop in the isotopes is generally interpreted as a drop in the temperature, as would naturally occur after a volcanic eruption. However, a concurrent drop in accumulation indicates a more complex dynamic, as the two proxies are likely to 685 be a correlated response to the atmospheric impact of volcanic eruptions. Accumulation changes may be caused by a change in the frequency of precipitation events, which is the leading mode of inter-annual variability and closely linked to the NAO by the position of the NA-storm track. As highlighted in Sjolte et al. (2018), there are indications of a positive NAO index after eruptions. A change in the seasonal distribution of precipitation may affect the $\delta^{18}$O independently from the temperature or possibly even counterbalancing any cooling.

These results show that the impact of eruptions on the Greenlandic ice sheet climate lasts, on average, for a decade after the eruptions, more than is usually estimated from global temperatures or from single-ice-core studies (Kobashi et al.,





2017). Nonetheless, the measured average decrease in layer thickness after the eruptions is too small to influence the wet/dry depositional effect on the measurement of aerosol impurities in the ice.




Figure 10 Left-most panels: signal averages around the 105 eruptions matched in the GICC21 timescale; time is flowing from right to left. Right panels: corresponding Monte-Carlo studies of the significance of the measurement. (a) The ECM average identifies a common feature in the shape of a sharp peak at 0 years, highlighting the common practice adopted in this work of marking the eruptions at the maximum rather than at the onset. The average duration of the peak is 3 years, starting 1 year before the maximum. (b.1) The $\delta^{18}O$ stack-average highlights a sharp decrease of the isotopes of about 0.12 ‰ (orange arrow), which starts about one year before the maximum of the ECM, at the onset of the ECM peak. A duration of about 10 years of the post-eruption minimum is observed (blue arrow). (b.2) The amplitude of the $\delta^{18}O$ decrease is compared to a Monte-Carlo simulation, highlighting the rather rare occurrence of such a decrease when compared to a random selection of 105 ages. (b.3) The duration of the cooling in the isotopes is compared to a Monte-Carlo




simulation. Again, the result seems to be rather rare compared to the random simulation. (c.1) The layer- thickness stack average shows a decrease of about 1 % with respect to the mean value and a duration of about 10 years, similar to the isotopic behavior. This indicates a drop in annual accumulation after an eruption, possibly an additional effect of the volcanic cooling. (c.2) The Monte-Carlo simulation returns a distribution of the amplitude drop that is Gaussian. The sample-value is placed in the right tail of the distribution at a significance distance from the mean. (c.3) The duration of the accumulation drop is also on the right side of the distribution, although the duration

measured by the algorithm is less than one would deduce from c.1 (i.e. 10 years). For a comparison of these results with GICC05, see Supplementary Information Fig. S 4.

### 5.3. Perspectives from the GICC21 timescale on Mediterranean eruptions

After having discussed the offset to IntCal20 and the average climate response on the Greenland ice sheet to eruptions, we want to show how ice-core data align with some actual examples of Mediterranean eruptions. So far, only two

Mediterranean eruptions were thought to have been found in Greenland in the last 4000 years. The Vesuvius eruption (79 CE) was certainly of large magnitude (VEI=5), but, since Mediterranean tephra in ice cores have not yet been found despite continuous sampling (Bourne et al., 2015; Cook, personal communication, 2021), hopes to find signals in Greenland have been scaled back. The second eruption is the debated Minoan eruption on Thera, Santorini. Although the eruption has an estimated VEI of 7 (Johnston et al., 2014), the expectations of finding direct evidence of tephra from Thera in ice cores are

rather low (McAneney & Baillie, 2019), because of the dominating circulation patterns that drive transport of volcanic signal eastward from the Mediterranean region. In the absence of tephra, by comparing Greenland acidity records and site-specific radiocarbon evidence and by considering that the offset to IntCal20 should have been resolved, we can only provide indirect evidence of the impact of Mediterranean eruptions at the very distant Greenlandic location.

In Sigl15, a composite of five northern Hemispheric tree-ring chronologies, called 'N-Tree', measures tree-growth anomalies

over the last 2500 years in the Northern Hemisphere. By looking at the alignment of ECM with the N-Tree record, we observe that Greenland eruptions align with low-growth years very accurately providing an indication of the timescale accuracy with respect to tree rings (Supplementary Information Fig. S 5). A comparison to another tree-ring growth reconstructions that reach until 3835 years b2k (Helama, Seppä, Bjune, & Birks, 2012) did not lead to conclusive evidence for a better alignment of GICC21 to growth minima with respect to GICC05, either because the resolution of the tree ring

data was too low or because no clear minima were seen in the vicinity of the ECM peaks.

A close observation of all ECM signals at 79 CE on the GICC21 age scale does not indicate that the Vesuvius eruption, although strong, left any prominent acidity signal on the ice sheet. As an additional comparison tool, we compared our data to the N-Tree composite from Sigl15, where we can observe the excellent alignment between the eruptions and the cooling in the Northern Hemisphere, providing more evidence for the accuracy of the GICC21 age model. We note that the year of

the Vesuvius eruption seems to align with a cooling both in the N-Tree record and in the isotopes, which by no means is conclusive evidence that the eruption was somehow registered in Greenland, but might indicate that there could be potential for further investigation, especially in the topic of ice-core crypto-tephra. Evidence found around Vesuvius includes a vase of olive oil, found in Pompeii, which was dated to 1944 ± 50 [14]C-years BP (Sacchi et al., 2020). The age of the sample was calibrated by converting the [14]C age of the sample according to IntCal20, through the OxCal v4 online tool

(Ramsey, 2009). If the age of Vesuvius was unknown, this comparison could provide an indication of the age, but the wide distribution of [14]C ages prevents an exact identification of the Vesuvius peak in the ice cores (Figure 11).



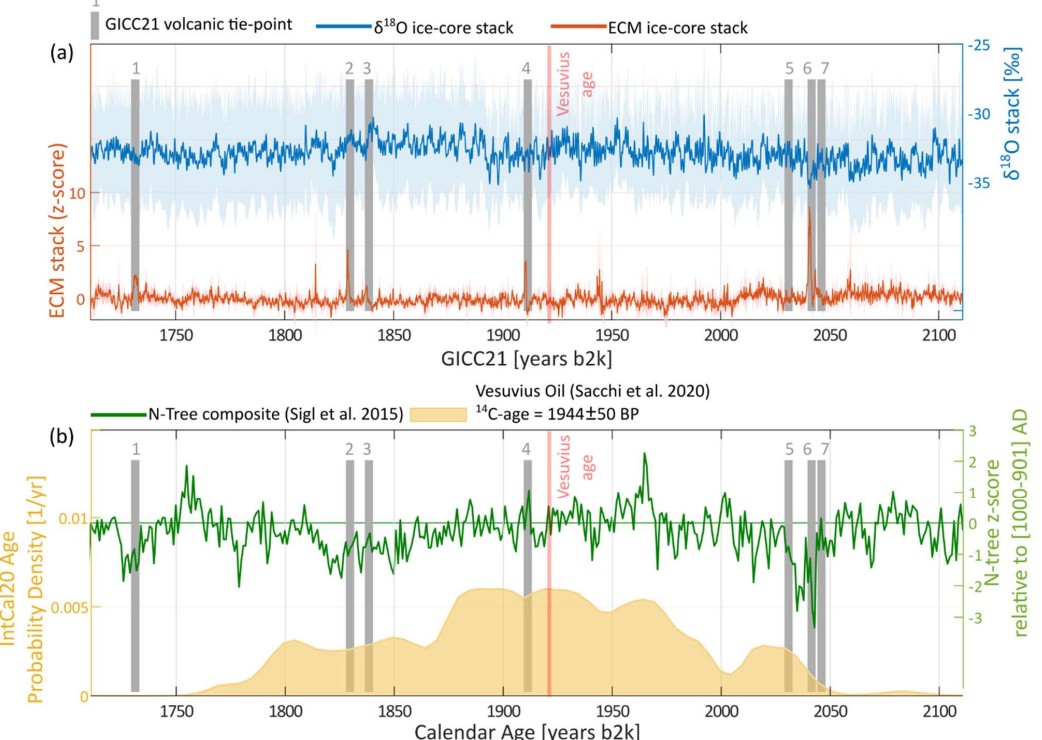

Figure 11 The red vertical bar marks where Vesuvius 79 CE should be found in GICC21, while the grey numbered bars indicate tie points used in GICC21. (a) No clear signal can be identified at the Vesuvius age in the ECM-stack of the ice cores and no identifiable peak was found in any of the single records. (b) In the years following the eruptions, marked cooling can be seen both in the isotope stack and in the N-Tree record of Sigl15, as shown in Supplementary Information Fig. S 5. It can be argued that some cooling in the N-tree record and the ice-core isotopes can be observed right after the Vesuvius age, however no clear causal relationship is demonstrated by this observation alone. The broad probability distribution of calibrated ages from the radiocarbon dated sample is not providing an independent age estimation for Vesuvius, of which we naturally know the true age.

To clarify whether the new timescale provides new perspectives on the Thera dating, we produce a comparison between ice core data and the Thera radiocarbon dated evidence (Figure 12). As discussed in Van der Plicht et al. (2020), the eruption unfortunately happened during a plateau in $^{14}$C production, which results in widening of dating uncertainties and a multi-modality of the age distribution. The outer layer of an olive branch found in Santorini and commonly associated with the eruption, which has a $^{14}$C age of 3331+/-10 years BP (Friedrich et al., 2006; Friedrich, Heinemeier, & Warburton, 2009), was modelled according to IntCal20 and shows multi-modality. However, by wiggle matching the olive tree rings to the IntCal20 dataset, the calendar dates for the outer layer are estimated to 3592-3612 years b2k (68.2% confidence) and 3578-3617 years b2k (95.4% confidence) (Van der Plicht et al., 2020).

The volcanic match in the Thera age-range – from 3500 to 3700 years b2k - relies on 6 volcanic tie points, of which the 5 youngest ones also correspond to NorthGRIP1 sulfate peaks. At these ages, only EastGRIP, NEEM, GRIP, and DYE-3 have data that allow annual-layer identification. According to the olive branch age-calibration of its outermost layer, the three eruptions older than 3620 years b2k are excluded from the picture. Specifically, the eruption labelled no.4 in Figure 12, now attributed to Aniakchak II (Alaska), was formerly attributed to Thera. The age of this eruption, according to the revised ice core chronology, becomes younger by 13 years, but is nonetheless outside of the main probability density of the radiocarbon dated olive branch. On the other hand, younger ECM peaks, traditionally disregarded as Thera candidates, are found to fall within the range of possible ages. Moreover, thanks to the new ice-core timescale, we found that the calcium anomaly in tree rings, reported by Pearson et al. (2020) in a sample from Turkey, aligns exceptionally well with a volcanic eruption visible in all deep ice cores at 3560 years b2k, labelled no.2 in Figure 12. However, tephra evidence for the origin of this volcanic eruption is still lacking.

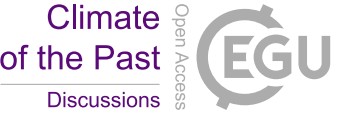

Since the Sigl15 N-Tree record ends 2500 years ago, we cannot use it to compare the tree-ring growth minima in the Thera age range. Instead, we referred to the bristlecone pine growth minima record compiled by Salzer & Hughes (2007), where we found very good correspondence between the 3626 and the 3649 years b2k growth minima of North-American trees and two volcanic tie points at those ages, the first of them being the Alaskan Aniakchak eruption (Pearce, Westgate, Preece, Eastwood, & Perkins, 2004), which again confirms the alignment of the eruptions in ice cores and tree-growth minima.

It is interesting to observe that, over the age intervals proposed by Van der Plicht et al. (2020) by wiggle-matching tree-ring data, the $\delta^{18}$O stack experiences a prolonged decrease, possibly indicating a cooling period. Also, a peak at 3610 years b2k (1610 BCE) is recorded in the ECM stack, right before the cooling. Inspection in the ice cores showed that the ECM peak is mostly visible in GRIP, while minor ECM enhancements can be seen in NEEM, and DYE-3 at the same age. The identity of this eruption has, to our knowledge, not yet been identified. The isotope decrease has a quantifiable duration of 9 years, starting at 3610 years b2k, and shows an isotope drop difference of 0.63±0.21 ‰ from the average measured in the 80
years before 3610 years b2k. On the other hand, the average layer thickness registers a jump of 0.033±0.028 but only a 3-year long minimum (not shown in Fig. 12, but available for reproduction in the Timescale Supplement). The drop in isotopes is certainly important, especially if compared to the histograms b.2 and b.3 in Figure 10, where it would be placed on the far-right end of the distribution tails. However, the layer thickness drop cannot be distinguished from a random fluctuation because of its short duration. Still, the values appear to be compatible with the volcanic cooling analysis we conducted in
5.2, indicating that this decrease in isotopes could, in principle, have been caused by a volcanic eruption.

In conclusion, multi-modality of the probability distribution and disagreement between radiocarbon ages from different samples prevents a unique identification of a candidate for the Thera eruption in the ice core ECM stack. No Mediterranean tephra has ever been found in a Greenland ice core despite continuous ice sampling, and the quest for the identification of the Santorini eruption remains open. However, a high-resolution search for crypto-tephra grains using scanning electron
microcopy (SEM) screening techniques, with particular focus on the isotope decrease at 3600 years b2k and on the peak at 3560 years b2k, might offer an answer to whether ashes from Santorini were deposited on the Greenlandic ice sheet. -----

------------------------------------------------------------

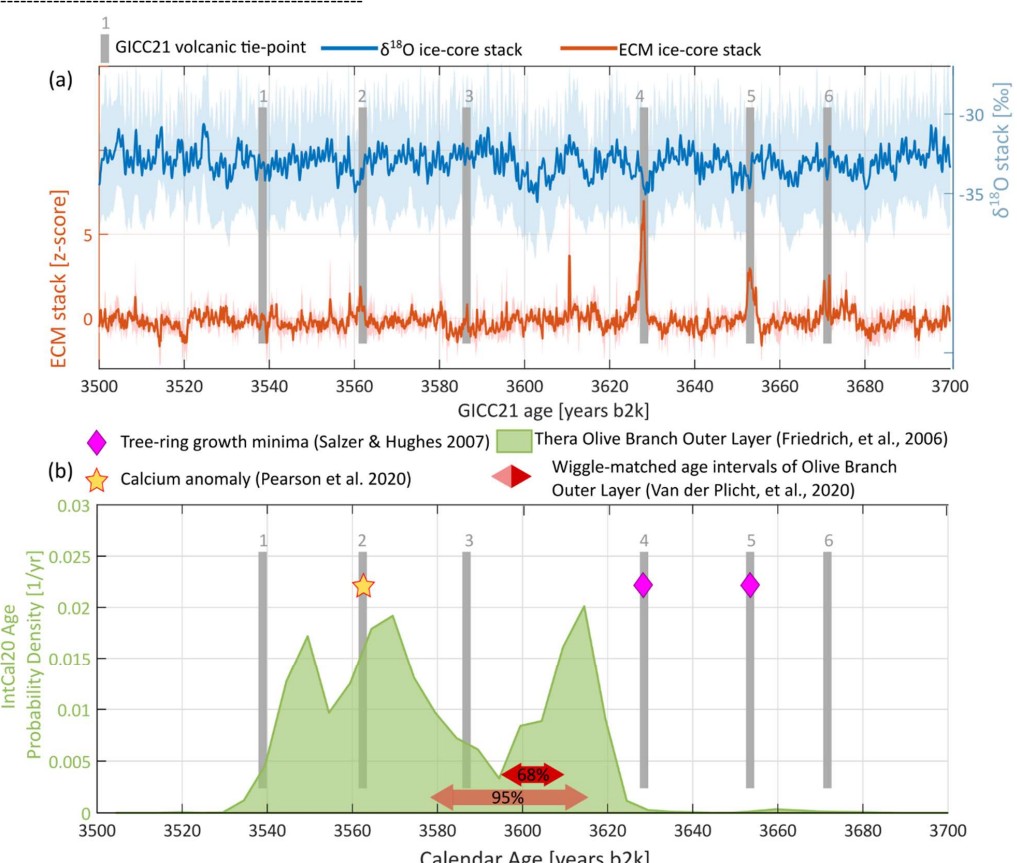



Figure 12 Comparison of ice-core data with the radiocarbon dated olive branch found in Santorini. (a) Stacked ECM and isotope record of EastGRIP, NEEM, GRIP, DYE-3 (the shading shows the standard deviation of the stack). Six volcanic matches are observed, corroborated by sulfate evidence in NorthGRIP. The ice cores show cooling after each eruption, except no.3, and a significant cooling period around 3600 years b2k. (b) The age probability density function of the olive branch ($^{14}$C-dated to 3331+/-10 years b2k) related to the Thera eruption. The numbered, grey bars highlight the synchronized volcanic tie-points used for GICC21, although uncertainty with respect to the alignment with IntCal20 should be considered in this timeframe. The three eruptions older than 3620 years b2k (no. 4, 5, and 6) don't overlap with the probability distribution and are therefore less likely to correspond with the Thera eruption, no.4 being Aniakchak. Two of them however overlap with two tree-growth minima (purple diamonds) reported by Salzer & Hughes (2007) in tree rings records from western USA, providing more support on the accuracy of the GICC21 timescale compared to tree-ring timescales. The red arrows highlight the possible Thera age range by Van der Plicht et al. (2020). It appears that this age range corresponds to the cooling seen in the ice core isotopes.

## 6. Conclusions

Compared to GICC05, the new GICC21 ice-core timescale shows higher potential for climatic studies and synchronization to distant climatic records in the late Holocene, such as radiocarbon dated evidence proximal to eruption sites. The timescale offset to GICC05 shows a non-linear behavior, as a consequence of local issues with the layer count and the ice-core comparison. Until 742 years b2k, the two timescales agree with uncertainties, which is very important for shallow ice-core studies. However, beyond the Samalas eruption (1258 CE, 742 years b2k), the offset increases rapidly because of the mismatch of both the Hekla (1104 CE) eruption and the Vesuvius (79 CE) eruptions.

The automated-counting algorithm StratiCounter was applied with success to recognize layers in the ice cores, using the new available proxy data from EastGRIP and NEEM, but nonetheless showed some intrinsic issues, since the algorithm was under-counting layers in some ice cores (EastGRIP, NEEM, and NEEM-S1-2011) and over-counting layers in others (GRIP, NorthGRIP, and especially DYE-3), as demonstrated in the well-constrained age range younger than the Samalas eruption. Hence, we demonstrated the need for a multi-observer manual fine-tuning and applied an empirical statistical approach to show that the rate of the timescale uncertainty envelope can be estimated as about 1 year per century, going back in time from Samalas. A lower bound of 2 years needs to be added to the uncertainty, to account for displacement and delays in the volcanic acidity deposition on the ice sheet. We remark that the existence of a counting bias in each ice core is not a failure of SC, as the task of recognizing layers is challenging regardless of the methodology applied. That is to say that the algorithm cannot overcome the bias which is as much an intrinsic problem with the annual layer record as it is an issue with the layer identification method. Since we demonstrated that ice cores do have site-specific disturbances that affect the layer count, it is clear that a multi-core comparison such as the one conducted in this work is essential to increase the accuracy of the timescale.

The timescale's offset from GICC05 reaches 13 years at 3835 years b2k, which is significant even considering the timescale uncertainty at this age. The offset has an oscillating behavior between 2000 years b2k and 3835 years b2k, with three important excursions from the mean at amplitude of about 5 years. This fact we attribute to matching issues related to widely-spaced volcanic eruptions, a finer ammonium-based match in GICC21, and layer thickness fluctuations in DYE-3.

The revision of the timescale was stopped at 3835 years b2k to ensure multi-core comparison. However, since NEEM data improves again for depths larger than 1200 m (corresponding to roughly 8 ka b2k), there is a possibility for a revised Early-Holocene ice-core chronology based on data from EastGRIP, NEEM, and DYE-3, made by a method similar to the one provided here. In contrast, between 3.8 and 8 ka b2k, i.e. within the typical brittle ice section of the cores where data quality is lower, any timescale revision will have to be constructed by different means or by acquisition of new data from new and old ice cores. In the meantime, we recommend a timescale-calibration offset toward younger ages when using GICC05 beyond 3835 years b2k (Appendix A).

Having obtained a very precise synchronization of six deep ice cores, we calculated the multi-core signal stacks at yearly resolution in order to study the common behavior of Greenland ice core proxies. The study focused on post-volcanic cooling, providing evidence for a decadal impact of eruptions on the regional climate. The registered decrease of isotopes (0.12 ± 0.02 ‰) and layer thickness (1.1 ± 0.3 %) after eruptions shows that the volcanic impact is significantly registered across Greenland, with substantial cooling and decrease of accumulation.

Comparison of GICC21 to previous studies provide new insight on some debated issues of ice-core timescales, such as the offset to the IntCal calibration curve. Thanks to the comparison of cosmogenic radionuclide records, we were able to compare IntCal20 and GICC21 in the Thera range, concluding that the offset is negligible within uncertainties back to at least 3300 years ago.

Comparison to tree-ring minima until 2500 years b2k suggests a very good alignment of the eruption signals in ice cores and the cooling registered across the Northern Hemisphere. Therefore, the alignment of two important tree-growth minima validates the new GICC21 chronology in the 17$^{th}$ century BCE. Inclusion of more tree-ring growth records into a composite



spanning back over the Holocene could provide better insight about the alignment of ice cores and tree rings growth, helping the validation of further ice-core timescale Holocene revisions.

Finally, evidence was provided that GICC21 aligns with radiocarbon dated evidence found at eruption sites, hinting to where crypto-tephra searches should be performed in the future, especially relating to the Thera eruption of Santorini. It is currently not possible to connect a specific acidity spike with this eruption, because of the multi-modality of the radiocarbon dated evidence, which indicates several possible ages. On one hand, evidence of draughts in Turkey (Pearson et al., 2020) correlate with a volcanic eruption at 3560 years b2k (1560 years BCE), co-registered in all Greenlandic ice cores but of which

the identity is still unknown. On the other hand, tree-ring data from an olive branch found on the Greek island was wiggle-matched to IntCal20 (Van der Plicht et al., 2020) indicating a narrower age-range that appears to align with a drastic cooling event in ice-core isotope data, at around 3600 years b2k (1600 years BCE). Both suggestions could be candidates for a crypto-tephra search in Greenlandic ice cores, as we remark that the ice core GICC21 timescale would allow a very precise dating of Thera with a tightly constrained uncertainty.

**Appendix A**

The transfer function was calculated by, first, acquiring the timescale offsets of each of the ice cores involved in this study. Then, we computed the uncertainty using equation (1). For each year between 0 and 3835 years b2k, a weighted mean of the offset and the corresponding weighted uncertainty is calculated. The transfer function is reported in the Timescale Supplement. For the ice cores EastGRIP, NEEM, NorthGRIP1, NorthGRIP2, and DYE-3 we advise the direct use of

the GICC21 layers and of equation (1) for the uncertainty, in order to convert ages from GICC05. For other cores, which were matched to one or a combination of these ice cores (e.g. GISP2, Rasmussen et al., 2008), we recommend the use of the average transfer function to translate ages from GICC05 to GICC21 until 3835 years b2k. For sections older than 3835 years b2k, we recommend ages of GICC05 to be calibrated by a shift towards younger ages. The amount of calibration needed is 14 years for DYE-3, 12 years for EastGRIP, 11 years for GRIP, and 12 years for any other ice core based on the

average transfer curve.

**Appendix B**

To be able to stack the signal of several ice cores at annual resolution, we first need to report each ice core's data on the correct annual average. Hence, for each ice core, we average the signal within each year, making sure that the data points overlapping with a layer boundary are weighted correctly across it. For example, if the annual layer boundary is 2 cm into

a 5 cm isotope sample, it will enter the annual averages on each side with 40% and 60%, respectively.

For each ice core, we set a threshold of 80% data coverage within each year, which means that if more than 20% of the data are missing during a certain year, we neglect that core in the averaging.

To do the stacking, we adopted proxy-specific normalizations. Because the calibration of the ECM is not accurate and since volcanic peaks have an highly variable amplitude between each cores, the ECM data are normalized by z-scores from the

millennial average. Fixed, consecutive windows were chosen as to preserve the relative amplitude of the volcaninc peaks over the slowly-varying background level.

The layer thickness data is nomalized by dividing by the centennial means, in order to partially account for thinning and firn densification. On the other hand, isotope data are kept on their original unit.

After normalizing, the data from different ice cores are aligned on the common GICC21 (or GICC05) timescale to compute

the mean and the variance of the stack. The variance is further divided by the number of ice cores involved in the stacking for each event to obtain the standard error of the stack.

**Appendix C**

The following Monte-Carlo protocol was used to test the significance of the isotope and layer thickness post-eruption decreases, by iterating the steps 10'000 times:

1. 105 random times were selected between 0 and 3835 years b2k. For the layer thickness only, in order to avoid firn densification effects on the amplitude, the most recent 100 years were excluded;
2. The stacks (isotopes and layer thickness) were selected for 100-years windows centered on each random time;
3. The 105 windows were averaged at annual resolution; a standard error was computed on the average signal;
4. The search for the amplitude and the duration was performed (described in the following).

The amplitude and duration are measured in the same way both for the randomized iterations and for the actual sample. The amplitude is computed as the difference between:



- $A_{before}$, the average signal in the 50 years before the eruption peak;
- $A_{after}$, the average signal in the 50 years after the eruption.

The duration of the signal drop is computed by, first, finding all clusters of consecutive data-points that are significantly below average, in a 30 year window around the eruption peak, and then selecting the longest one in terms of duration. This represents a conservative test since the actual measurement has to compete with all the longest minima in the ensemble.

## 7. Data availability

### 7.1. The timescale layers and Straticounter pre-processing steps

The timescale layer boundaries, the volcanic/$NH_4^+$ tie-points, and the GICC05-GICC21 transfer function are available in the Excel supplement provided with the online version of the paper. The stacks of layer thickness, isotope data, and ECM are also made available in the same document.

The preprocessing steps required to run StratiCounter are available as a separate Excel supplement with the online version of the paper.

### 7.2. Data

Data underlying the timescale, if not yet published, will be made available in three separate publications by:

1. Rasmussen et al. (in prep.) will contain data from DYE-3, NGRIP, GRIP, NEEM used for GICC05 and GICC21.
2. Erhardt et al (2021, in review) contains the CFA data from NEEM and NGRIP, which is available on PANGEA under https://doi.pangaea.de/10.1594/PANGAEA.935838 (dataset in review).
3. Erhardt et al. (in prep.) will contain CFA data from EastGRIP.

While these publications are on their way, preliminary data files can be obtained from the authors.

## 8. Supplementary Information

Supplementary Information is available as a separate document at the online version of the paper.

## 9. Author contributions

G.S. drafted the paper with comments and corrections form all co-authors. G.S. produced the SC raw counts and led the fine-tuning, which was performed together with M.W. and S.O.R, who also provided guidance on methodology. S.O.R. produced the manual gap count for the uncertainty study, supervised the general content of the work, and provided ideas regarding the uncertainty formula and the Monte-Carlo study.

B.V. produced the correlation study for DYE-3, GRIP, and SWG temperatures. A.S. provided suggestions for the IntCal comparison and the tree-ring minima comparison. T.E. and C.J. provided information about the EastGRIP data. C.J. helped 925 with research on ammonium signal in ice cores. T.E. provided useful insight and ideas about the uncertainty and the statistical framework. E.C. provided information about tephra evidence in the Holocene. R.M produced the single realizations of the IntCal20 [14]C production rates by carbon-modelling, as well as providing guidance for the IntCal comparison.

## 10. Conflicts of interest

The authors declare that they have no conflict of interest.

## 11. Acknowledgements

### 11.1. Funding

G.S. and S.O.R. acknowledge support via the ChronoClimate project funded by the Carlsberg Foundation.

T.E. and C.M.J. acknowledge the long-term support of ice core research at the University of Bern by the Swiss National 935 Science Foundation (SNSF) (#20Fi21_164190) and the Oeschger Center for Climate Change Research.

EastGRIP is directed and organized by the Centre for Ice and Climate at the Niels Bohr Institute, University of Copenhagen. It is supported by funding agencies and institutions in Denmark (A. P. Møller Foundation, University of Copenhagen), the USA (US National Science Foundation, Office of Polar Programs), Germany (AlfredWegener Institute, Helmholtz Centre for Polar and Marine Research), Japan (National Institute of Polar Research and Arctic Challenge for





Sustainability), Norway (University of Bergen and Trond Mohn Foundation), Switzerland (Swiss National Science
        Foundation), France (French Polar Institute Paul-Émile Victor, Institute for Geosciences and Environmental Research),
        Canada (University of Manitoba) and China (Chinese Academy of Sciences and Beijing Normal University).

        **11.2. People**

        G.S. thanks Hubertus Fischer for his useful comments to the manuscript.

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
