# Peer review of "A multi-ice-core, annual-layer-counted Greenland ice-core chronology for the last 3800 years: GICC21"

_Climate of the Past, 2021_

## Community Comment (CC1)

**Review Sinnl et al.: A multi-ice-core, annual-layer-counted Greenland ice-core chronology for the last 3800 years: GICC21**

The authors provide a revision of the previous GICC05 age-scale for Greenland ice cores for the past 3,800 years. They combine automated layer recognition approaches with synchronization techniques to develop a dating framework which is both consistent between ice cores and annually dated. This tedious work is of great importance not only for ice-core science but also welcomed in paleoclimatology. Overall, I think the GICC21 timescale is a great improvement of the GICC05 and the previous long standing dating bias has been corrected for this time period. The stacked records of ECM, layer thickness and $\delta^{18}O$ are very useful paleo-climate proxies with much increased signal-to-noise properties as accessible from individual ice cores. They are used to discuss two time windows suspected to contain the volcanic eruptions of Vesuvius 79 CE and the Minoan eruption of Thera.

Below I provide a list of detailed comments to clarify some uncertainties which are aimed to improve the quality of the presentation of the results. It has been 16 years after GICC05 was constructed for this revision so I assume no additional revision is planned anytime soon for this time period. Thus the documentation of this timescale should be as comprehensive, clear and accurate as possible. Before that I would like to summarize four more general points:

**A) Input data for annual-layer counting**

Ideally, such an effort would include all available suitable ice cores. As presented, it appears to me that you have incorporated only the CFA data from NEEM analyses done in the field, but you have not incorporated the CFA-ICPMS data from the same core analyzed in the trace chemistry lab at the Desert Research Institute on the NEEM Steering Committee piece (abbreviated with NEEM-SC in the SOM file of Sigl et al., 2015). This dataset is published for the time period 500 BCE to 146 CE. Since you here use SC as an abbreviation for StratiCounter I wonder if you may have assumed that the file NEEM-SC describes the StratiCounter results of the traditional NEEM-CFA analyses. It is instead a completely independent analysis on a parallel NEEM ice-core section with state-of-the-art analytical instrumentation. The NS1-2011 timescale is solely based on manual layer counting using the two NEEM aerosol datasets (see Figure 2 for a comparison of the calcium data) as reported by Sigl et al. (2015), though the co-author M. Winstrup may have done some StratiCounter analyses on the data. Clarification is thus needed what the NEEM-SC COUNT (Figure 6) actually represents. It is probably too late now to include the NEEM data from DRI into an already existing dating framework, but please point out that you haven't considered all available ice-core analyses, and that there are differences in the input data used between GICC21 and NS1-2011.

**B) Absolute age markers**

Absolute age markers -- when correctly identified and well dated -- have the potential to improve the accuracy and reduce the uncertainty of an annual-layer dated chronology. However, if they are ambiguous and in cases erroneous, they may also contribute to manifest in a dating bias (e.g., as was demonstrated on the examples of Vesuvius 79 and Hekla 1104; or the infamous 1453 Kuwae eruption to which ice-core chronologies from Antarctica had been tuned for many years). Finding the best balance is difficult and can be subjective. For NS1-2011, I have been trying to use minimal age constraints (1258, 939, 775, 626, 536) and let StratiCounter define the annual layers in between. I noted that key volcanic age markers discussed in the literature are exactly reproduced by GICC21, such as 536, 626 and 939 CE (Sigl et al., 2015); Okmok II in 43 BCE (McConnell et al., 2020) and Aniakchak II in 1629 BCE (Cole-Dai et

al., 2021). Since overall, there are quite some age differences between the different ice cores, I wonder if these ages have been prescribed when constructing the chronology?

On the other side, it appears that the [10]Be anomalies identified by Sigl et al., (2015) which anchor the NS1-2011 chronology (see Figure 1) have not been used to constrain GICC21 for those ice cores for which this data exist (NEEM-2011-S1, NGRIP). Specifically, if I transfer the GICC21 ages to the only available annually dated [10]Be data (i.e. cutting the samples along the annual-layer boundaries) from NEEM-2011-S1, it appears that the [10]Be rise had started in 773 (see Figure 1), a year before the SPE event was supposed to have occurred in boreal summer 774 (Buntgen et al., 2018). A shift of GICC21 +1 year (consistent with NS1-2011) would also bring inline a sharp and short-lived sulfate spike in 800 CE with a short-lived cooling recorded by tree-rings in 800 CE (Sigl et al., 2015).

[Figure]

**Figure 1:** (left): NEEM-2011-S1 annual [10]Be concentrations on the NS1-2011 chronology (Sigl et al., 2015) versus stacked annual tree-ring [14]C content from a northern hemisphere tree-ring network. Shaded area marks the timing an uncertainty of the SPE event in boreal summer 774 CE (Büntgen et al., 2018). Right: the same data plotted on the GICC21 chronology.

Another potential and frequently used age marker is radioactive fallout from nuclear weapon testing (peaking 1954-1963) which has been identified for some of the ice cores used in GICC21 (Arienzo et al., 2016). Have such age markers been considered for GICC21? This might also be used to validate the potential to use NH4 for age synchronization, because there are prominent biomass burning signals in Greenland ice cores in 1961 and 1964 (Legrand et al., 2016; McConnell et al., 2007). A precise and accurate chronology is in particular valuable for the most recent period due to the overlap with observations, reanalysis and remote sensing.

**C) Relative age markers: volcanoes versus biomass burning**

For the alignment of the ice-cores in between the constraints of the 105 major volcanic eruptions you make extensive use of $NH_4$ (some 240 major tie-points according to the SOM) which is among other sources often emitted during biomass burning. The low number of widely accepted volcanic tie-points and the high number of $NH_4$ tie-points, which to my knowledge hasn't been used before to align ice cores across the Greenland ice sheet, are the biggest surprises in the manuscript. I wouldn't go so far to reject the idea that many or even the majority of the $NH_4$ tie-points are correctly tying together the corresponding biomass burning episodes, at least for ice cores located closely to each other, e.g. NGRIP and EastGRIP. But I don't find the rational and supporting data convincingly laid out in this manuscript. If biomass burning events indeed left a unique fingerprint in the $NH_4$ throughout Antarctica, you should be

able to validate this using independent biomass burning tracers (such as black carbon) not used for synchronization. There is a continuous 2,500 year record of black carbon available from the NEEM-2011-S1 and NEEM ice cores (Sigl et al., 2013; Sigl et al., 2015) cited by (Zennaro et al., 2014). Can you confirm in the NEEM ice cores the biomass burning source for your major $NH_4$ tie-points by co-registration of BC peaks, considering that black carbon is in the pre-industrial atmosphere a unique fire tracer?

Whereas black carbon records are numerous in Antarctica they have never been used for chronological synchronization purposes owing to the large spatial and temporal variability of black carbon over Antarctica (Liu et al., 2021; McConnell et al., 2021). This doesn't mean per se that it couldn't be possible in Greenland but the burden of proof lies on the side of the authors.

**D) Thera eruption**

In advent of ongoing developments in the radiocarbon community and with the revised dating and stacking of relevant proxy records at hand, new insights may indeed be forthcoming; however, we need to be careful to not over-interpret the results. A minimum in stable isotope values in Greenland may have many explanations, including that of a major volcanic eruption. But in the absence of evidence of a clear stratospheric eruption signal in the ice cores (i.e. increased sulfuric acidity deposition over several years uniformly over Greenland) a volcanic source is the least likely explanation for the assumed cooling, let alone that is was specifically the Thera eruption. The Thera eruption has already previously been tied to a major cooling event in 1627 BCE, which subsequently was revised. This period interesting and worth reporting but I would try to emphasize the speculative nature of the current appraisal of the evidence.

**Specific comments:**

**Abstract:**

**L. 4 & l.10:** Please be specific which ice-core chronology was revised (i.e. GICC05); there are several annual-layer counted chronologies for ice cores from Greenland notably from GISP2 (Meese-Sowers), NEEM-2011-S1 (NS1-2011) and NGRIP2 (DRI_NGRIP2). The latter two do not have a dating bias, and the one from GISP2 has not been assessed in your paper.

**L. 16:** The statement that "cooling lasted for up to a decade, longer than reported in previous studies of volcanic forcing" is incorrect. There is a large body of literature reporting that volcanic cooling may have lasted for up to a decade (Büntgen et al., 2020; Büntgen et al., 2016; Sigl et al., 2015; Tejedor et al., 2021), some of which you are citing in l.37-38.

**L. 20-21**: This is speculation. There is no evidence of a major stratospheric volcanic eruption at the time in any ice core so this stable isotope anomaly is most likely not related to volcanic activity. If at all this speculation belongs into the discussion rather than in the abstract.

**Main text:**

**L. 25-26:** I would believe this statement is controversially discussed in the field of human history and difficult to prove either wrong or possible.

**L. 59:** $Na^+$ instead of $Na^{2+}$ (happened a few times throughout the paper)

**L. 59-63:** How are the seasons of the impurity peaks inferred?

**L. 68-68:** Counting can also been done on insoluble parameters.

**L. 80-81:** Whereas the use of sulfate for age synchronization is common in the field, the use of biomass burning events has to my knowledge never been used for this purpose, unless for ice cores drilled a few meters apart. Maybe this could be better highlighted by citing relevant literature.

**L. 80-87:** the wording is a bit imprecise in this section: with sulfate you can't "locate individual eruptions" but rather identify eruption signals in the ice strata; the duration of volcanic sulfate deposition is highly variable and depends on many factors, there are numerous sulfate depositions lasting less than a year, though these signals are usually not used for volcanic synchronization purposes, because they are not equally widespread and uniformly deposited over the Greenland ice sheet. I would suggest replacing "event" with "deposition signal" to make clear that you don't refer to the start and end of the eruptions which you also called events in l.80.

**L. 89-90:** Consider adding the relevant tephra papers which overlap with the age range of GICC21, the last 3,800 years. I haven't found any mention of tephra in the methods and results sections, except a link to the Laki 1783 eruption. Have you been using the available tephra evidence to constrain the absolute dating (e.g. Veiðivötn Feb. 1477; Tianchi, Nov. 946) or relative dating (e.g., Churchill 853; Okmok II 43 BCE) of the ice cores in GICC21? If none of the tephra has been used to constrain or evaluate the GICC21 timescale, why mention it in the introduction of the paper?

**L. 105-107:** the literature research on the ability of ammonium or other biomass burning proxies in ice core for synchronizing ice cores across the Greenland ice sheet is incomprehensive. There is a vast body of literature analyzing the extent to which biomass burning left unambiguous, reproducible signals between different proxies (e.g. ammonium, black carbon, vanillic acid) and/or between different ice core locations in Greenland or Antarctica (Keegan et al., 2014; Legrand et al., 2016; Liu et al., 2021; McConnell et al., 2007; Zdanowicz et al., 2018). Legrand et al. (2016) for example only identified 9 biomass burning events with synchronous NH4 deposition in a 200 year-window using three precisely dated high-accumulation ice cores.

**L. 110-115:** Have these events been used for the GICC21 chronology? If so, it would be valuable to provide a table with the ages and corresponding depths for each of the ice cores in which 10Be anomalies have been identified. I have been plotting the only available annual 10Be data from the NEEM-2011-S1 ice core (Sigl et al., 2015) on the GICC21 chronology and note that the 10Be rise occurred in 773 CE a year before the solar proton event has occurred.

**L. 131:** Incorrect ice core stated here: "we identify tephra particles and determine that volcanic shards extracted from a depth of 429.3m in the GRIP ice core are likely due to the 79AD Vesuvius eruption". (Barbante et al., 2013)

**L. 132:** Very vague statement. Recent geochemical analyses of tephra shards from the NEEM-2011-S1 directly associated with the large acidity peak demonstrate that the peak is not linked to the Vesuvius eruption but points to other potential sources including from Alaska. The tephra from NEEM-2011-S1 is geochemically distinct from the shards described by Barbante at al. (2013) which have no direct stratigraphic context with the acid peak, but appear a year earlier. Note that the cited Discussions paper is now in press.

**L. 133:** I think the synchronization has never been an issue here. The sulfate peak is large and clear. It is the association of the peak with a historic event and the subsequent transfer of its supposed age and uncertainty into the GICC05 framework which were causing an issue.

**L. 134:** The issue of a potential age bias in the GICC05 chronology has also been raised by e.g. (Baillie, 2008, 2010; Lohne et al., 2014, 2013; Torbenson et al., 2015)

**L. 137-138:** This is incorrect and needs to be corrected. Coulter et al. (2012) was targeting the supposed Hekla 1104 period in three ice cores (Dye-3, GRIP, NGRIP). No tephra was found in Dye-3, and GRIP but four shards were identified in NGRIP (QUB-1186). However, these are situated c. four years before the onset of the massive sulfate peak which had been previously attributed to Hekla 1104 eruption in GICC05. We have also targeted this sulfur peak in the NEEM-2011-S1 and TUNU2013 ice cores (using large cross sections) and have not found any tephra, as summarized by Plunkett et al., (in press). Consequently, the NS1-2011 (Sigl et al., 2015) and DRI_NGRIP2 (McConnell et al., 2018) annual-layer counted chronologies have not been constrained by this erroneous match to the historic Hekla 1104 eruption. Moreover, no sulfur isotope measurements have been reported from any Greenland ice core in the literature. A recent study focusing on this eruption signals in ice cores has suggested that multiple eruptions, including an eruption of Asama in 1108 may have contributed to the distinctive signal in the Greenland ice cores (Guillet et al., 2020).

**L. 161-163:** To clarify, the new NS1-2011 chronology (Sigl et al., 2015) constructed for the NEEM-2011-S1 ice core is between 1258 CE and 2013 CE identical with the previous age model for this ice core which was constrained by volcanic eruption dates taken from GICC05 and annual-layer counting in between these marker years. Before 1258 CE, the NS1-2011 chronology was no longer constrained by GICC05 including the previous matches to Vesuvius 79 CE and Hekla 1104 CE, but was instead constrained by three well-dated observations of volcanic dust veils from documentary sources (i.e. 536, 626 and 939 CE) and the solar proton events of 774 and 993 CE. Between these marker events, StratiCounter was used to identify annual-layer boundaries in the multi-parameter impurity records. Other than Samalas, no bipolar tie-point is employed to constrain NS1-2011 before 1258 CE.

**L. 161-163:** Sigl et al., (2013) manually interpreted annual layers in NEEM-2011-S1 in between the prescribed GICC05 dated volcanic markers. Sigl et al. (2015) used StratiCounter on the same NEEM-2011-S1 dataset after replacing the GICC05 dated volcanic markers with the new tie-points as described above.

**L. 164-165:** To clarify, the 15 volcanic match-points between NGRIP and NEEM referred to here are only for the time period 88 CE until 500 BCE. For the time period for which NGRIP sulfate data was available (190-1999 CE) we synchronized the NGRIP sulfate data to the NS1-2011 chronology using 123 volcanic tie-points identified in NGRIP and NEEM-2011-S1.

**L. 166:** The number of years counted in the aerosol records from the two independent NEEM core analyses was c. 590 years. How about "were conducted until 500 BCE"?

**L. 182/83:** Are you sure you used all available data from Greenland deep ice cores? Based on section 2.3, you used only the measurements done on the main NEEM core in the field by traditional CFA, but you omitted the data from the trace element analyses done on NEEM between 500 BCE and 146 CE at the Desert Research Institute, which is the data mainly used for the NS1-2011 chronology before 88 CE. Analyzed with two high-res. mass spectrometers in a class 100 clean room, this data is ideally suited to resolve intra-annual variations in multiple impurities records (e.g. Sigl et al., 2016; McConnell et al., 2018) and less likely to be contaminated during data acquisition compared to the CFA field analyses (see

Figure 2, which I want to emphasize is not representative for the entire NEEM CFA analyses). Also the deep GISP2 ice core has been dated by annual-layer counting.

[Figure]

*Figure 2:* *Comparison of $Ca^{2+}$ concentrations (in ppb) by CFA analyses in the field (Ca Swiss, red) with Ca concentrations (in ppb) determined by ICPMS under controlled clean room conditions (Ca DRI, black) on the NEEM ice core for a 4 m section dated 17-38 CE (Sigl et al., 2015). Note that this section is not representative for the entire NEEM CFA analyses; it is used to highlight the quality of available impurity records not used for GICC21.*

**L. 214-215:** In which ice cores was tephra found? The Coulter et al. (2012) reference is missing here describing the tephra results from NGRIP. Trace elements confirming the link to Thera are presented in Plunkett et al. (2017).

**L. 233-251:** Your description of the datasets is a bit imbalanced. You could elaborate a little on the NorthGRIP2 dataset, which just like EastGRIP include a vast range of species measured continuously at high resolution, but at a site with almost twice the accumulation rate of EastGRIP. Since GICC21 closely follows the independent annual-layer counted DRI_NGRIP2 chronology (McConnell et al., 2018) for most part, it seems reasonable to assume that this new record is the cornerstone of the new chronology.

**L. 333-340:** I am still missing when you lay out your arguments (ideally backed up with data and/or relevant publications) why you assume that $NH_4$ peaks are synchronous across Greenland with a frequency of 0.1 years$^{-1}$. Are these $NH_4$ peaks thought to be from individual biomass burning plumes? Is there observational evidence for such widespread deposition? For volcanic eruptions the long atmospheric lifetime (especially following stratospheric eruptions) provides very distinctive sulfate signals regarding duration, magnitude and shape. Therefore, their use is well established in the ice-core dating community. For $NH_4$, a similar framework is still missing, and the data as presented in the paper doesn't yet convince me that the additional use of $NH_4$ will be better than a tight volcanic alignment with annual-layer counting in between. In the past 2,500 years, the frequency of volcanic eruptions detectable in Greenland ice cores is 0.09 years$^{-1}$ (Sigl et al., 2015).

**L. 380-381:** Necessarily, if you aim for a unified, synchronized chronology for several cores, subjective decisions need to be made. And these subjective decisions can become influenced by prior knowledge that we have. For example, on what we know about the dating bias towards too old ages in GICC05 throughout the Holocene. The age offsets between ice-core 10Be and tree-ring $^{14}$C are well established. Frost-ring events potentially linked to volcanic eruptions have been proposed in the last years. Can you comment on steps taken to ensure such prior information has not influenced the manual fine tuning?

**L. 383-84:** I would have limited this exercise to volcanic eruptions for which tephra was clearly identified in ice cores, therefore to my knowledge excluding Hekla 1510. Typo for the 1362 eruption. How did you identify the Oræfajökull 1362 eruption in the EastGRIP, NorthGRIP, NEEM and NEEM-2011-S1 ice cores? There is hardly any sulfate peak in many ice cores, nor was the tephra identified in the GRIP ice core (Coulter et al. 2012) associated with an ECM peak.

**L. 471:** Do you argue that the uncertainty in GICC21 is never below 2 years?

**L. 519:** Why is your frequency of volcanic tie-points so low compared to the frequency of eruptions in Sigl et al. (2015), 1 every 11 years?

**L. 544:** Figure 6: It seems that the comparison of GICC05 and NS1-2011 has been omitted between 87 CE and 187 CE creating an apparent gap in the data, which doesn't exist. Is this correct? GICC05 ages of NEEM-2011-S1 are available by Sigl et al., (2013) and the corresponding NS1-2011 ages are published in Sigl et al., 2015 together with a transfer function between NS1-2011 and GICC05. https://www.nature.com/articles/nature14565#Sec24

**L. 548:** Please add reference for the NS1-2011 chronology (Sigl et al., 2015).

**L. 553:** Wouldn't a potentially wrong $NH_4$ tie-point also produce a drift between the ice cores?

**L. 591:** There are no 130-year long periods without any volcanic eruption.  To clarify: Do you mean you don't see a volcanic eruption in any of the cores or a volcanic eruption signal which can be detected in all of the cores? Using established methods and the NGRIP2 chemistry data (McConnell et al., 2018), volcanic eruptions can be detected on average every 13 years, with the longest repose period lasting less <60 years.

**L. 598-99:** This may be true, but it doesn't mean per se that several ice cores are always better than a single ice core (as the previous age bias in the GICC05 ice cores has shown). Note, there is no offset between the GICC21 and DRI_NGRIP2 chronology for almost 1,500 years (i.e. between 200 BCE and 1270 CE, Figure 3) despite the latter being based on a single ice core (NGRIP2), and only four of the annual-layer decisions in GICC21 deviate from those from the previous annual-layer-counting age model (McConnell et al., 2018). The observed offset is more likely attributable to the reduced core quality of NEEM-2011-S1 is some sections around 300 CE.

[Figure]

*Figure 3: Age difference between the annual-layer counted DRI_NGRIP2 chronology (McConnell et al., 2018) and GICC21 between 700 and 2200 yb2k.*

**L. 661:** better to write following all selected (or all 105) eruptions. The number of eruptions detectable in Greenland is much larger (i.e. Sigl et al., 2015 detected 221 eruptions in Greenland in only 2,500 years).

**L. 674:** Have you accounted in your stacked analyses for the fact that volcanic eruptions often formed temporal clusters (e.g. 536/540, 1453/1459, 1809/1815), and that an apparent persistency (i.e. your secondary dip) may be an artifact caused by secondary major eruptions? In tree-ring research on the volcanic climate response, this has been taken into account (Büntgen et al., 2020). In Plunkett et al., (in press) we demonstrate that a decadal scale reduction of d18O from Dye-3, GRIP and NGRIP is explained by the cooling response following a cluster of eruptions dated 536, 540 and 547 CE.

**L. 717:** I am not sure one can cite a personal communication from a co-author. The most comprehensive and recent review article of tephra deposits in the Holocene in Greenland is (Plunkett and Pilcher, 2018) which also addresses the situation for tephra from the Mediterranean.

**L. 724:** In 2015, no annually dated Northern Hemispheric temperature reconstruction was available, thus we created the N-Tree composite. In the meantime, more comprehensive reconstructions have become available and have already been used to demonstrate the accuracy of the NS1-2011 chronology in the Common Era (Büntgen et al., 2020). Since NS1-2011 and GICC21 are close to each other, I would assume that these results would look the same for GICC21.

**L. 726:** eruptions identified in Greenland ice cores.

**L.731:** Have you analyzed relevant sections from any of the ice cores encompassing 79 CE for the presence of tephra?

**L. 737:** I miss the relevance of the radiocarbon dating of material in context with the exactly dated 79 CE Vesuvius eruption for the problem of identifying volcanic fallout from this eruption in Greenland. I suggest that the point would be better removed, to bring the focus more back to the ice core chronology.

**L. 762:** Please provide relevant references here.

**L. 767:** Here and again later (e.g l. 854) you refer to the 3560 year b2k ice-core anomaly. In the supplementary data the rise of the ECM signal is in 3562 year b2k, and the peak in 3561 year b2k. Can you please clarify the age of the eruption signal in GICC21?

**L. 773:** I think it would be appropriate to cite in this section the pioneering work of (Baillie, 2010) who noted a number of the mismatches between tree-ring anomalies and ice-core acidity peaks which now appear to have been corrected.

**L. 785:** If the assumed 10-year cooling would have been caused by a volcanic eruption it would have been through the stratospheric sulfate aerosols. The sharp and short-lived (1 year only) ECM spike characterized by strong spatial variations in Greenland is more likely the result of a tropospheric eruption rather than a major stratospheric eruption. There is no credible evidence in the ECM data and available sulfate records from Greenland (GISP2, (Zielinski et al., 1996)) around 1610 BCE of any significant stratospheric sulfur injection. Linking the drop in $\delta^{18}O$ to a volcanic eruption is thus only speculation. Internal variability, changes in atmospheric circulation and moisture sources and/or solar activity are just as likely candidates to explain the d18O feature.

**L. 788:** It would be helpful for future research if you provided (if possible) metadata on which ice cores and which cross sections ($cm^2$) have been sampled continuously in this time window. To my understanding, NGRIP ice core has been fairly extensively studied by Coulter et al., (2012); GRIP was studied by Hammer et al (1987). NEEM, EastGRIP may have been studied by Eliza Cook, cited before as

personal communications and this paper would greatly benefit from more specific detail about the nature of any tephra work contributing to GICC21. The new timescale revision will certainly stimulate new studies around this time period, and it would be helpful to know the state-of-the-art regarding tephra investigations in this critical period.

**L. 818:** This additional ±2 year uncertainty could be reduced for a number of events for which (1) atmospheric transport times are short (<weeks), (2) the timing of tephra arrival is well documented by high-time resolution particle records and (3) independent age constraints (e.g. dendrochronological, documentary or tree-ring evidence) can provide sub-seasonal information. I would argue that Veiðivötn (Feb. 1477 (Abbott et al., 2021)), Tianchi (November 946 (Oppenheimer et al., 2017) whose tephra in Greenland is reported by Sun et al. 2014) and the solar proton events in 993 CE, 774 CE and 660 BCE can all be dated with uncertainties less than ±2 years, and the list could be expanded including Eldgjá 939, unidentified volcanic eruptions in 626, 536 CE (Sigl et al., 2015) and the eruption of Okmok II in winter 44/43 BCE (McConnell et al., 2020). I have, however, no objections if you are leaning towards a more conservative approach regarding the uncertainties for these events.

**L. 823:** A multi-ice core comparison is favorable rather than essential (see Figure 3). I would further argue that the single ice-core WD2014 chronology is more accurate than the multi ice-core GICC05 chronology over the Holocene (Sigl et al., 2016). It is the quality of the expression of the intra-annual variations that matters most, not the number of ice cores.

**L. 853:** Typo: Droughts.

**L. 911:** All datasets used for the timescale should be made publicly available at high depth resolution to allow for independent validation of the timescale. This was not possible for GICC05, but is now mandatory under FAIR open data principles.

**References:**

Abbott, P. M., Plunkett, G., Corona, C., Chellman, N. J., McConnell, J. R., Pilcher, J. R., Stoffel, M., and Sigl, M.: Cryptotephra from the Icelandic Veiðivötn 1477 CE eruption in a Greenland ice core: confirming the dating of volcanic events in the 1450s CE and assessing the eruption's climatic impact, *Clim. Past*, 17, 565-585, 2021.

Arienzo, M. M., McConnell, J. R., Chellman, N., Criscitiello, A. S., Curran, M., Fritzsche, D., Kipfstuhl, S., Mulvaney, R., Nolan, M., Opel, T., Sigl, M., and Steffensen, J. P.: A Method for Continuous (PU)-P-239 Determinations in Arctic and Antarctic Ice Cores, *Environ Sci Technol*, 50, 7066-7073, 2016.

Baillie, M. G. L.: Proposed re-dating of the European ice core chronology by seven years prior to the 7th century AD, *Geophys Res Lett*, 35, 2008.

Baillie, M. G. L.: Volcanoes, ice-cores and tree-rings: one story or two?, *Antiquity*, 84, 202-215, 2010.

Barbante, C., Kehrwald, N. M., Marianelli, P., Vinther, B. M., Steffensen, J. P., Cozzi, G., Hammer, C. U., Clausen, H. B., and Siggaard-Andersen, M. L.: Greenland ice core evidence of the 79 AD Vesuvius eruption, *Clim Past*, 9, 1221-1232, 2013.

Büntgen, U., Arseneault, D., Boucher, É., Churakova, O. V., Gennaretti, F., Crivellaro, A., Hughes, M. K., Kirdyanov, A. V., Klippel, L., Krusic, P. J., Linderholm, H. W., Ljungqvist, F. C., Ludescher, J., McCormick,

M., Myglan, V. S., Nicolussi, K., Piermattei, A., Oppenheimer, C., Reinig, F., Sigl, M., Vaganov, E. A., and Esper, J.: Prominent role of volcanism in Common Era climate variability and human history, *Dendrochronologia*, 64, 125757, 2020.

Büntgen, U., Myglan, V. S., Ljungqvist, F. C., McCormick, M., Di Cosmo, N., Sigl, M., Jungclaus, J., Wagner, S., Krusic, P. J., Esper, J., Kaplan, J. O., de Vaan, M. A. C., Luterbacher, J., Wacker, L., Tegel, W., and Kirdyanov, A. V.: Cooling and societal change during the Late Antique Little Ice Age from 536 to around 660 AD, *Nat Geosci*, 9, 231-236, 2016.

Büntgen, U., Wacker, L., Galvan, J. D., Arnold, S., Arseneault, D., Baillie, M., Beer, J., Bernabei, M., Bleicher, N., Boswijk, G., Brauning, A., Carrer, M., Ljungqvist, F. C., Cherubini, P., Christl, M., Christie, D. A., Clark, P. W., Cook, E. R., D'Arrigo, R., Davi, N., Eggertsson, O., Esper, J., Fowler, A. M., Gedalof, Z., Gennaretti, F., Griessinger, J., Grissino-Mayer, H., Grudd, H., Gunnarson, B. E., Hantemirov, R., Herzig, F., Hessl, A., Heussner, K. U., Jull, A. J. T., Kukarskih, V., Kirdyanov, A., Kolar, T., Krusic, P. J., Kyncl, T., Lara, A., LeQuesne, C., Linderholm, H. W., Loader, N. J., Luckman, B., Miyake, F., Myglan, V. S., Nicolussi, K., Oppenheimer, C., Palmer, J., Panyushkina, I., Pederson, N., Rybnicek, M., Schweingruber, F. H., Seim, A., Sigl, M., Churakova, O., Speer, J. H., Synal, H. A., Tegel, W., Treydte, K., Villalba, R., Wiles, G., Wilson, R., Winship, L. J., Wunder, J., Yang, B., and Young, G. H. F.: Tree rings reveal globally coherent signature of cosmogenic radiocarbon events in 774 and 993 CE, *Nat Commun*, 9, 2018.

Cole-Dai, J., Ferris, D. G., Kennedy, J. A., Sigl, M., McConnell, J. R., Fudge, T. J., Geng, L., Maselli, O. J., Taylor, K. C., and Souney, J. M.: Comprehensive Record of Volcanic Eruptions in the Holocene (11,000 years) From the WAIS Divide, Antarctica Ice Core, *Journal of Geophysical Research: Atmospheres*, 126, e2020JD032855, 2021.

Guillet, S., Corona, C., Ludlow, F., Oppenheimer, C., and Stoffel, M.: Climatic and societal impacts of a "forgotten" cluster of volcanic eruptions in 1108-1110 CE, *Sci Rep-Uk*, 10, 10, 2020.

Keegan, K. M., Albert, M. R., McConnell, J. R., and Baker, I.: Climate change and forest fires synergistically drive widespread melt events of the Greenland Ice Sheet, *P Natl Acad Sci USA*, 111, 7964-7967, 2014.

Legrand, M., McConnell, J., Fischer, H., Wolff, E. W., Preunkert, S., Arienzo, M., Chellman, N., Leuenberger, D., Maselli, O., Place, P., Sigl, M., Schüpbach, S., and Flannigan, M.: Boreal fire records in Northern Hemisphere ice cores: a review, *Clim Past*, 12, 2016.

Liu, P. F., Kaplan, J. O., Mickley, L. J., Li, Y., Chellman, N. J., Arienzo, M. M., Kodros, J. K., Pierce, J. R., Sigl, M., Freitag, J., Mulvaney, R., Curran, M. A. J., and McConnell, J. R.: Improved estimates of preindustrial biomass burning reduce the magnitude of aerosol climate forcing in the Southern Hemisphere, *Science Advances*, 7, 10, 2021.

Lohne, O. S., Mangerud, J., and Birks, H. H.: IntCal13 calibrated ages of the Vedde and Saksunarvatn ashes and the Younger Dryas boundaries from Krakenes, western Norway, *J Quaternary Sci*, 29, 506-507, 2014.

Lohne, O. S., Mangerud, J., and Birks, H. H.: Precise C-14 ages of the Vedde and Saksunarvatn ashes and the Younger Dryas boundaries from western Norway and their comparison with the Greenland Ice Core (GICC05) chronology, *J Quaternary Sci*, 28, 490-500, 2013.

McConnell, J. R., Chellman, N. J., Mulvaney, R., Eckhardt, S., Stohl, A., Plunkett, G., Kipfstuhl, S., Freitag, J., Isaksson, E., Gleason, K. E., Brugger, S. O., McWethy, D. B., Abram, N. J., Liu, P., and Aristarain, A. J.:

Hemispheric black carbon increase after the 13th-century Māori arrival in New Zealand, *Nature*, 598, 82-85, 2021.

McConnell, J. R., Edwards, R., Kok, G. L., Flanner, M. G., Zender, C. S., Saltzman, E. S., Banta, J. R., Pasteris, D. R., Carter, M. M., and Kahl, J. D. W.: 20th-century industrial black carbon emissions altered arctic climate forcing, *Science*, 317, 1381-1384, 2007.

McConnell, J. R., Sigl, M., Plunkett, G., Burke, A., Kim, W. M., Raible, C. C., Wilson, A. I., Manning, J. G., Ludlow, F., Chellman, N. J., Innes, H. M., Yang, Z., Larsen, J. F., Schaefer, J. R., Kipfstuhl, S., Mojtabavi, S., Wilhelms, F., Opel, T., Meyer, H., and Steffensen, J. P.: Extreme climate after massive eruption of Alaska's Okmok volcano in 43 BCE and effects on the late Roman Republic and Ptolemaic Kingdom, *P Natl Acad Sci USA*, 117, 15443-15449, 2020.

Oppenheimer, C., Wacker, L., Xu, J., Galvan, J. D., Stoffel, M., Guillet, S., Corona, C., Sigl, M., Di Cosmo, N., Hajdas, I., Pan, B., Breuker, R., Schneider, L., Esper, J., Fei, J., Hammond, J. O. S., and Büntgen, U.: Multi-proxy dating the 'Millennium Eruption' of Changbaishan to late 946 CE, *Quaternary Sci Rev*, 158, 164-171, 2017.

Plunkett, G., Pearce, N.J., McConnell, J., Pilcher, J., Sigl, M. & Zhao, H.: Trace element analysis of Late Holocene tephras from Greenland ice cores. *Quaternary Newsletter* 143, 10–21, 2017.

Plunkett, G. and Pilcher, J. R.: Defining the potential source region of volcanic ash in northwest Europe during the Mid- to Late Holocene, *Earth-Sci Rev*, 179, 20-37, 2018.

Plunkett, G., Sigl, M., Schwaiger, H., Tomlinson, E., Toohey, M., McConnell, J. R., Pilcher, J. R., Hasegawa, T., and Siebe, C.: No evidence for tephra in Greenland from the historic eruption of Vesuvius in 79 CE: Implications for geochronology and paleoclimatology, *Clim. Past*, in press.

Sigl, M., Fudge, T. J., Winstrup, M., Cole-Dai, J., Ferris, D., McConnell, J. R., Taylor, K. C., Welten, K. C., Woodruff, T. E., Adolphi, F., Bisiaux, M., Brook, E. J., Buizert, C., Caffee, M. W., Dunbar, N. W., Edwards, R., Geng, L., Iverson, N., Koffman, B., Layman, L., Maselli, O. J., McGwire, K., Muscheler, R., Nishiizumi, K., Pasteris, D. R., Rhodes, R. H., and Sowers, T. A.: The WAIS Divide deep ice core WD2014 chronology - Part 2: Annual-layer counting (0-31 ka BP), *Clim Past*, 12, 769-786, 2016.

Sigl, M., McConnell, J. R., Layman, L., Maselli, O., McGwire, K., Pasteris, D., Dahl-Jensen, D., Steffensen, J. P., Vinther, B., Edwards, R., Mulvaney, R., and Kipfstuhl, S.: A new bipolar ice core record of volcanism from WAIS Divide and NEEM and implications for climate forcing of the last 2000 years, *J Geophys Res-Atmos*, 118, 1151-1169, 2013.

Sigl, M., Winstrup, M., McConnell, J. R., Welten, K. C., Plunkett, G., Ludlow, F., Büntgen, U., Caffee, M., Chellman, N., Dahl-Jensen, D., Fischer, H., Kipfstuhl, S., Kostick, C., Maselli, O. J., Mekhaldi, F., Mulvaney, R., Muscheler, R., Pasteris, D. R., Pilcher, J. R., Salzer, M., Schüpbach, S., Steffensen, J. P., Vinther, B. M., and Woodruff, T. E.: Timing and climate forcing of volcanic eruptions for the past 2,500 years, *Nature*, 523, 543-549, 2015.

Sun, C. Q., Plunkett, G., Liu, J. Q., Zhao, H. L., Sigl, M., McConnell, J. R., Pilcher, J. R., Vinther, B., Steffensen, J. P., and Hall, V.: Ash from Changbaishan Millennium eruption recorded in Greenland ice: Implications for determining the eruption's timing and impact, *Geophys Res Lett*, 41, 694-701, 2014.

Tejedor, E., Steiger, N., Smerdon, J. E., Serrano-Notivoli, R., and Vuille, M.: Global Temperature Responses to Large Tropical Volcanic Eruptions in Paleo Data Assimilation Products and Climate Model Simulations Over the Last Millennium, *Paleoceanography and Paleoclimatology*, 36, e2020PA004128, 2021.

Torbenson, M. C. A., Plunkett, G., Brown, D. M., Pilcher, J. R., and Leuschner, H. H.: Asynchrony in key Holocene chronologies: Evidence from Irish bog pines, *Geology*, 43, 799-802, 2015.

Zdanowicz, C. M., Proemse, B. C., Edwards, R., Feiteng, W., Hogan, C. M., Kinnard, C., and Fisher, D.: Historical black carbon deposition in the Canadian High Arctic: a 250-year long ice-core record from Devon Island, *Atmos. Chem. Phys.*, 18, 12345-12361, 2018.

Zennaro, P., Kehrwald, N., McConnell, J. R., Schüpbach, S., Maselli, O. J., Marlon, J., Vallelonga, P., Leuenberger, D., Zangrando, R., Spolaor, A., Borrotti, M., Barbaro, E., Gambaro, A., and Barbante, C.: Fire in ice: two millennia of boreal forest fire history from the Greenland NEEM ice core, *Clim Past*, 10, 1905-1924, 2014.

Zielinski, G. A., Mayewski, P. A., Meeker, L. D., Whitlow, S., and Twickler, M. S.: A 110,000-yr record of explosive volcanism from the GISP2 (Greenland) ice core, *Quaternary Res*, 45, 109-118, 1996.

---

## Author Comment (AC2)

[Figure]

Figure 1 Example of the co-occurrence of $NH_4^+$ and NO3 peaks in the NEEM ice cores.

[Figure]

Figure 2 Alignment of log-inverted ECM (pseudo-$NH_4^+$) peaks and true $NH_4^+$ peaks in the NEEM ice core.

---

## Author Comment (AC5)

[Figure]

*Figure 1 NEEM-2011-S1 Black Carbon and NH₄ comparison. The blue bars are not GICC21 tie-points but merely indicate the co-occurrence of NH₄ peaks with BC peaks.*

---

## Author Response (AR1)

Response and Changes made to the Manuscript of

"A multi-ice-core, annual-layer-counted Greenland ice-core chronology for the last 3800 years: GICC21"

We made sure to address all comments from the reviewers, from the more technical ones to the more complex ones.

**Abstract**

The abstract was shortened as a consequence of removing sections about volcanic cooling and the Thera eruption of Santorini.

**Introduction**

We aimed at improving the introduction to be more informative and comprehensive.

We added the suggested references throughout the introduction.

We added a short description of differences in dating methods, as suggested by RC1.

In Section 1.3. Holocene stratigraphic markers, we added more references about using ammonium peaks for ice core comparison.

In Section 1.4.1, Uncertainty estimates of GICC05 in the Holocene, we provided more information on the GICC05 MCE and the problem of bias.

In Section 1.5. The NS1-2011 timescale, we tried to improve our brief review of the ns1-2011 timescale.

In Section 1.6., The need for a revised and unified Greenland ice-core chronology in the Holocene, we weakened our statement about using all data form Greenland.

**Data**

We added more information on the resolution of the datasets. We also improved our ice-core data availability section at the end of the MS.

**Methods**

We created a new supplementary table of the relevant Straticounter settings. The pre-processing of the data was explained more.

We verified the volcanic tie points of GISP2 and included it in our tephra review of Section 4 (where we added a new table of chronostratigraphic markers). We also revised our fine tuning with the new NEEM-SC data by Sigl et al., 2015 and found no issues with our layer counting in this section. We also used black carbon from NEEM and NEEM2011S1 to revise our ammonium match.

The ammonium matching procedure has been explained in a new section added ad hoc. Figure 2 was improved with the aim of highlighting ammonium as a matching tool. We quantified how the ammonium tie points are distributed in the timescale in a dedicated supplementary figure (s10). We used black carbon from NEEM and NEEM2011S1 to revise our ammonium match in order to limit our matching to the tie points that have both ammonium and BC, where possible.

We desist from calling the log-inverted ECM 'pseudo-NH4' as we recognize this name to be misleading.

We clarify the use of Laki as the datum of GICC05 and we add more explanation on the bce/ce conversion to years b2k, which is also added in the timescale supplement.

We aimed at clarifying where we expect we could be biased in our dating process and what precautions we took to avoid it.

Section 3.4 about the correlation study of DYE-3 was moved to the supplementary information to shorten the manuscript even more and to keep a better focus on our methodology.

Section 3.5 (now 3.4), Uncertainty of the GICC21 chronology, was shortened and rewritten to provide a more compact explanation of our uncertainty formula. We removed the redundant tables and figures and presented the new Figure 3 to visualize our statistical tests. We give more explanations about how we reach our formula. We conducted our uncertainty analysis in a continuous fashion, as suggested by RC2. The numerical analysis is reported in the SC supplement and some content was moved to the supplementary information.

**Section 4 (Results and Discussion)**

Figure 6(now 4) was changed following suggestion by RC2 by splitting it in two panels. We added the GISP2 timescale (Meese et al., 1997) and the DRI_NGRIP2 timescale (McConnell et al., 2018) in our timescale comparison.

We added more information on the DYE-3 400 ka b2k offset in the current figure 6.

We added more information on the timescale comparison and added current table 4 to summarize the chronostratigraphic markers most important for this study and their age according to GICC21.

We edited the section about the comparison to IntCal (now 4.3) by including some of the evidence found in the now-removed section about Thera, e.g. the comparison to tree ring data.

We removed the paragraphs about volcanic cooling and Thera, as we recognize these topics to be out of the scope of the present publication.

**Conclusions and Appendix**

We shortened this section in response to the removal of the mentioned paragraphs.

**Supplement**

We verified the Timescale Supplement to contain all relevant information about GICC21. We added a document about the StratiCounter pre-processing.

**Ice-core data availability**

All data is potentially available to the reader and we describe how to access it. We are encountering delays with Pangaea, hence the data being handled by SOR has been submitted but is not yet available. We reiterate that all datasets can be obtained via communication with SOR or TE.

Kind regards,

Sinnl et al.

---

## Author Response (AR2)

Dear Prof. Eric Wolff,

Thank you for your decision of accepting our manuscript into Climate of the Past.

We have corrected your comments as follows:

>>Line 55 - you cite Vinther06, but you don't actually explain your use of this abbreviation until line 65. To be honest I don't understand why you use this notation - it saves only 11 characters each time, and seems to be an unnecessary exception. I would recommend removing it (similar with Sigl15).
Reply: Removed Vinther06 and Sigl15 throughout the manuscript.

>>Lines 130-133: the logic is missing here. If the NEEM tephra is different from the GRIP one, and the NEEM one is Alaskan, this doesn't prove that the GRIP one (used in GICC05 I assume) is not Vesuvius. I accept that it isn't but please rewrite this sentence to explain more clearly what the issue is.
Reply: We added the point of Plunkett et al also criticising the GRIP tephra attribution to Vesuvius to strengthen the logical argument.

Line 151. Agreement of GISP2 with GICC is "rather poor". Please define this better eg "with mismatches of up to x years at nnnn years b2k".
Reply: We added the reference to Svensson et al., 2008 since they provide wider discussion of the topic of GISP2-GICC05 agreement. We added that GISP2 is -40 years at 8000 years b2k.

Line 168. Since you no longer think the peak is Vesuvius, this is a confusing wording. I suggest "the part younger than the eruption peak previously assigned to Vesuvius".
Reply: We have made the edit.

Table 2. There is an error in line 1. You say this is Laki - Hekla which would have an N of 261. If N is 306 then I guess you mean Laki-Barda. Please correct.
Reply: We have corrected to Barda.

Line 417. "we tested the correlation between Dye 3 and GRIP" - of what? I assume isotopes? Please say this.
Reply: We have specified it's isotopes.

Fig 3. Should we see the error, delta-t as a 1 sigma? It says so in the sup spreadsheet at sheet 1, section 2, but not in the text as far as I can see. Please clarify.
Reply: We specified that dt should be taken as 1σ.

Line 692: "The transfer function". Do you mean "The transfer function between GICC05 and GICC21"?
Reply: We have made the edit.

References in text: you occasionally gibe two authors followed by et al. This is an unusual format and I think incorrect. Normally its "Smith and Jones 2000" but if 3 authors "Smith et al 2000". Please check this.
Reply: We went through the citations and found some cases of 3+ authors being listed, which were changed to a format like "Smith et al.". We verified that only one author was cited in the other citations, and that only citations with 2 authors had both names "Smith & Jones".

Supplementary spreadsheet sheet 5 (transfer function). Anyone who picks this up and hasn't carefully read the paper won't know that delta-t is the uncertainty (as it's not a very common terminology). I can

see it stated n sheet 1 under section 2, but would prefer to see it stated as an extra row on sheet 5 (or at minimum under section 5 on sheet 1 as well).
Reply: We added a column description at the top of the sheet, referring to the uncertainty formula in the paper.

Supp spreadsheet. At first I looked in vain for the actual depth-age data, ie the key data from the paper. It took me quite a while to realise it was hiding under the obscure title "layer boundaries". Could you rename this tab to "depth vs age for GICC2021" to direct the reader to the right sheet easily please.
Reply: Yes, we agree. We have changed the sheet name and the header in the description to "Depth vs. Age GICC21"

I have decided to accept your data availability statement even if it isn't very satisfying for readers of this paper. Please be sure this paper is clearly referred to for all the datasets when they are finally released at Pangaea. I would also encourage you to put the ECM data listed as being available at CIC onto Pangaea.
Reply: Thank you for accepting our data comment. The paper will be referred in Pangaea. We will aim at reproducing the CIC ECM data also in Pangaea in the future.

Kind regards,
Sinnl et al.